# Superparameterised cloud effects in the GCM EMAC (v2.50) - influences of model configuration

**Harald Rybka**[1,2] **and Holger Tost**[1]

[1]Institute for Atmospheric Physics, Johannes Gutenberg-University, Mainz, Germany
[2]German Weather Service, Offenbach am Main, Germany

**Correspondence:** Harald Rybka (harald.rybka@dwd.de) and Holger Tost (tosth@uni-mainz.de)

**Abstract.** A new module has been implemented in the ECHAM5/MESSy Atmospheric Chemistry (EMAC) Model that simulates cloud related processes on a much smaller grid. This so called superparametrisation acts as a replacement for the convection parametrisation and large-scale cloud scheme. The concept of embedding a cloud resolving model (CRM) inside of each grid box of a general circulation model leads to an explicit representation of cloud dynamics. The new model component is evaluated against observations and the conventional usage of EMAC using a convection parametrisation. In particular, effects of applying different configurations of the superparametrisation are analysed in a systematical way. Consequences of changing the CRMs orientation, cell size and number of cells range from regional differences in cloud amount up to global impacts on precipitation distribution and its variability. For some edge case setups the analysed climate state of superparametrised simulations even deteriorates from the mean observed energy budget.

In the current model configuration different climate regimes can be formed, that are mainly driven by some of the parameters of the CRM. Presently, the simulated total cloud cover is at the lower edge of the CMIP5 model ensemble. However, certain "tuning" of the current model configuration could improve the slightly underestimated cloud cover, which will result in a shift of the simulated climate.

The simulation results show that especially tropical precipitation is better represented with the superparameterisation in the EMAC model configuration. Furthermore, the diurnal cycle of precipitation is heavily affected by the choice of the CRM parameters. However, despite an improvement of the representation of the continental diurnal cycle in some configurations, other parameter choices result in a deterioration compared to the reference simulation using a conventional convection parameterisation.

The ability of the superparametrisation to represent latent and sensible heat flux climatology is independent of the chosen CRM setup. Evaluation of in-atmosphere cloud amounts depending on the chosen CRM setup shows that cloud development can significantly be influenced on the large-scale using a too small CRM domain size. Therefore, a careful selection of the CRM setup is recommended using 32 or more CRM cells to compensate for computational expenses.

## 1 Introduction

Cloud related processes are difficult to simulate on the coarse grid of a general circulation model (GCM) and have a substantial influence on the global climate (Boucher et al., 2013). Small-scale effects like deep convection need to be parametrised in global models uncovering the problem that Earth System Models (ESMs) horizontal grid spacing requires further refinement to resolve cloud formation. Uncertainties in different atmospheric fields are primarily a consequence of using parametrisations (Zhang and McFarlane, 1995; Knutti et al., 2002), which rely on a physical basis but are mostly scale dependent including an arbitrary number of simplifications and assumptions. Nowadays, computational capabilities are suitable to perform global or large-domain simulations with resolution on the order of a few kilometres (Kajikawa et al., 2016; Heinze et al., 2017a) or even sub-kilometer grid spacing (Miyamoto et al., 2013). Convective-permitting simulations have shown that these models are able to realistically represent the Madden-Julian oscillation (MJO) (Miura et al., 2007; Miyakawa et al., 2014), the diurnal cycle of precipitation (Sato et al., 2009; Yashiro et al., 2016) or the monsoon onset (Kajikawa et al., 2015). Re-

solving the total effects of small-scale atmospheric features can hardly be simulated by any GCM with parameterised physics. The dilemma with these global cloud-resolving models (GCRMs) is the simulation period, that is limited by the computational expense to a couple of months nowadays. On that account coarser horizontal resolutions are necessary regarding long-term simulations, e.g. climate projections. A pioneer high-resolution (14 km global mesh) multi-year climate simulations has been conducted by Kodama et al. (2015). In addition to that, the first coordinated long-term model intercomparison of high-resolution (at least 50 km grid-size) climate simulations is underway within the High Resolution Model Intercomparison Project (HighResMIP) (Haarsma et al., 2016) of the Coupled Model Intercomparison Project 6 (CMIP6) (Eyring et al., 2016). The former examples showed that current developments and models still use resolutions that require a convection parametrisation in order to investigate climate related questions. Combining the ability to reproduce small-scale cloud dynamics by a cloud-resolving model (CRM) and perform long-term simulations with a GCM resulted in the idea of a „superparameterisation" (Grabowski and Smolarkiewicz, 1999; Grabowski, 2001; Khairoutdinov and Randall, 2001).

The concept of the superparametrisation is based on embedding a CRM inside of each column of the GCM replacing convection and large-scale cloud parametrisations. The superparametrisation acts as a conventional parametrisation but in contrast explicitly resolving small-scale cloud dynamics on the subgrid-scale of the GCM with the exception of cloud microphysics and turbulence. The CRM domain involves periodic lateral boundary conditions and forcings of large-scale tendencies, computed by the GCM, that are applied horizontally uniform. Finally, all small-scale effects represented by the mean of all CRM columns within one GCM grid-box interact with larger-scale atmosphere circulations on the coarse grid of the host model. Consequently, no direct interactions between individual CRM cells across GCM grid boundaries are possible. The computational cost of performing simulations with this framework is drastically reduced in contrast to a fully global cloud-resolving model (Grabowski, 2016). Including a CRM for the representation of the multitude of different types of clouds is a major step toward a more realistic representation of individual clouds and their interactions that are otherwise only achievable with high resolution models over huge domains.

After the first implementation of the superparametrisation several other institutes have followed the same approach (Subramanian et al., 2017; Tulich, 2015; Tao et al., 2009) and others are under way (Arakawa et al., 2011). Nowadays state-of-the-art global cloud resolving models provide new possibilities, comparing superparameterised simulations with monthly-long high resolution models (Stevens et al., 2019). In addition to this, the Center for Multiscale Modeling of Atmospheric Processes (CMMAP) has been created as a National Science Foundation's Science and Technology Center extensively progressing the work with superparametrisations (Randall et al., 2003; Randall, 2013; Khairoutdinov and Randall, 2006; Wyant et al., 2009; Elliott et al., 2016). Diverse modifications exist, which incorporate other processes or schemes within the embedded small-scale model, like a two-moment microphysical scheme (Morrison et al., 2009), a higher order turbulence closure or including aerosol coupling (Gustafson et al., 2008; Cheng and Xu, 2013; Wang et al., 2011a, b; Minghuai et al., 2015). These studies have mainly focused on improving selected process descriptions within the cloud-resolving model. This study presents an additional superparametrised GCM, primarily focusing on the effects of different CRM model configurations onto the mean climate state. Multiple simulations spanning 15 months have been performed to statistically evaluate the effects of changing different aspects of the superparametrisation, i.e. orientation, grid spacing and cell number of the embedded CRM. To our knowledge this is the first attempt summarizing the effects of different configurations of the superparametrisation onto the model mean climate state.

This paper is organized as follows. Section 2 describes the host GCM and CRM that are used as the superparametrisation. Furthermore, the coupling between the two model systems and the simulation setup is given. Section 3 examines the results of the new model system and discusses the sensitivity study comparing different superparametrised model setups. Section 4 gives a summary and conclusions.

## 2 Model Description

### 2.1 EMAC model system

Historically speaking, the ECHAM/MESSy atmospheric chemistry (EMAC) model (Joeckel et al., 2010) combines the Modular Earth Submodel System (MESSy) framework with the fifth generation of the ECMWF/Hamburg (ECHAM5) climate model (Roeckner et al., 2006). Developments during the last decade have fully modularised the code into the different layers of MESSy (Joeckel et al., 2005) and split representations of atmospheric processes into their own submodels. Based on that, alternative process descriptions (e.g. convection parametrisations, Tost et al., 2006) and even diverse base models (e.g. Community Earth System Model (CESM, Baumgaertner et al., 2016) or the COSMO model, Kerkweg and Jöckel, 2012) can be easily selected and compared for sensitivity climate simulations. EMAC has been used for various scientific applications regarding chemistry climate interactions from the surface to the mesosphere[1]. A complete list of available submodels is given in Table 1 in Joeckel et al., 2010.

---

[1]see http://www.messy-interface.org/ for a recently updated list of publications featuring MESSy

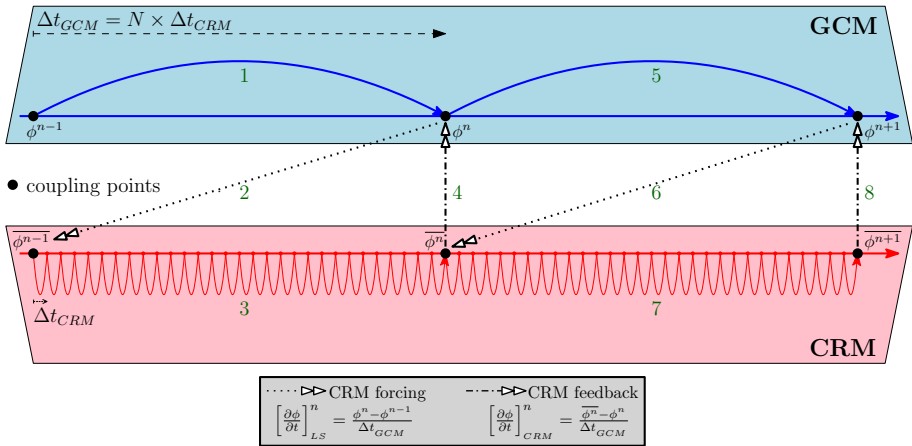

**Figure 1.** Sketch of model time integration when coupling a host GCM with a superparametrisation (i.e. CRM) based on their prognostic variables $\phi$ over a period of two time steps ($n$). A description of the different phases (i.e. numbers) is given in the text.

## 2.2 New submodule: CRM

As mentioned in the introduction a CRM has been implemented as a new submodel to serve as a superparametrisation (SP) for EMAC. The new coupled model system is therefore shortly named SP-EMAC. The CRM component of SP-EMAC is the System for Atmospheric Modeling (SAM; described in Khairoutdinov and Randall, 2003) that describes subgrid-scale development of moist physics in each GCM grid column. It solves the nonhydrostatic dynamical equations with the anelastic approximation. The prognostic variables are the liquid/ice water moist static energy, total precipitating water (rain + snow + graupel) and non-precipitating water (vapor + cloud water + cloud ice). An „all-or-nothing" approach is used to diagnose cloud condensate assuming saturation with respect to water/ice. The hydrometeor partitioning is based on a temperature dependence using a single moment microphysical scheme with fixed autoconversion rates. Additional information on the CRM is described in more detail in the Appendix of Khairoutdinov and Randall (2003).

The model code of the superparametrisation has been restructured to follow the MESSy coding standards. Thereby it is now possible to set specific parameters via namelist entries in order to obtain the flexibility for sensitivity analysis without recompiling the code. The main switches that can be adjusted change the configuration of the superparametrisation, i.e.:

- number of CRM grid cells inside of each GCM grid box

- grid size of CRM cells

- orientation of the CRM columns (2D or 3D)

- top height of CRM grid box

- time step of the superparametrisation

Each grid column of the global model EMAC hosts several copies of the CRM. All configurations of the superparametrisation use periodical lateral boundaries and a time step of 20 seconds. Vertical levels (29 in total) are aligned to match the lowermost levels of the GCM. Newtonian damping is applied to all prognostic variables in the upper third of the grid to reduce gravity wave reflection and build up. Communication between CRM cells across GCM boundaries is done via large-scale tendencies thereby neglecting direct interactions of small-scale dynamics between coarse grid columns.

## 2.3 SP-EMAC: Coupling the two model systems

Combining EMAC and the superparametrisation is based on applying the CRM forcing and CRM feedback of prognostic variables $\phi$ between the two models. But first and foremost vertical profiles of the coarse grid cells of EMAC are initialized in all CRM columns at the beginning of each model run. Simultaneously small temperature perturbations are added for near surface layers to obtain an individual response for each CRM column. During the simulation the CRM is called on every GCM time step and repeatedly integrating its equations while saving all subgrid-scale fields of the superparametrisation at the end of the call. A sketch of the GCM-CRM coupling is given in figure 1 displaying two GCM time steps ($n$) and their sequential phases during the model time integration. The numbers in figure 1 correspond to the following actions:

1 $\rightarrow$ integration of GCM (time step of $\Delta t_{GCM}$)

2 $\rightarrow$ coupling: CRM large-scale forcing $\left[\frac{\partial \phi}{\partial t}\right]_{LS}$

3 $\rightarrow$ integration of CRM ($N$-times $\Delta t_{CRM}$)

4 $\rightarrow$ coupling: CRM small-scale feedback $\left[\frac{\partial \phi}{\partial t}\right]_{CRM}$

The large-scale forcing $\left[\frac{\partial \phi}{\partial t}\right]_{LS}^{n}$ restricts the superparametrisation close to the host model fields whereas CRM feedback $\left[\frac{\partial \phi}{\partial t}\right]_{CRM}^{n}$ tendencies are calculated by the horizontal mean of all CRM grid boxes ($\overline{\phi^n}$) for timestep $n$ (Grabowski, 2004b). Prognostic variables that have been coupled are: temperature and moisture in terms of water vapor, cloud water and cloud ice. Horizontal momentum transport was only allowed for the 3D CRM configurations applying the same relaxation terms (see figure 1) to the $u$ and $v$ wind field components. All 2D CRM configurations neglect zonal/meridional convective momentum on the large-scale flow, i.e. no CRM feedback.

With regards to the computation of cloud optical properties and radiative fluxes two possibilities exist.

1. calculate radiative transfer with averaged cloud properties assuming a maximum-random overlap assumption obtained by averaging over the superparametrisation domain.

2. calculate radiative transfer explicitly with time-averaged CRM fields in every subgrid-scale column.

In this paper only the first possibility is chosen although including explicit cloud inhomogeneities into radiative transfer computation have a significant influence on radiative fluxes (Cole et al., 2005). The capability to consider subgrid-scale cloud-radiation interactions have been introduced after performing sensitivity simulations and will therefore not be part of the evaluation in this paper. With regard to cloud-radiation coupling all SP-EMAC simulation are equivalent to experiment 3 of Cole et al. (2005).

Further coupling is not implemented in the superparametrised version of EMAC so far. All land surface fluxes and boundary layer processes are simulated on the large-scale grid only (Roeckner et al., 2003). Surface heterogeneities like soil moisture, soil type, orography etc. may be included for future research with SP-EMAC. Large-scale moist physics are turned off when selecting the superparameterisation. Advection of mean (average over all CRM cells within a GCM column) prognostic fields (moisture, temperature, etc.) is enabled, whereas cloud cover and precipitation are diagnostics. It is assumed that sedimentation of all hydrometeors occurs within a GCM time step.

## 2.4 Simulation Setup

All simulations are performed with a horizontal GCM resolution of T42 and 31 vertical hybrid pressure levels up to 10.0 hPa. The applied setup for the control simulation (CTRL) covers the submodels for radiation (Dietmüller et al., 2016), clouds (Lohmann and Roeckner, 1996) and convection (Tiedtke, 1989) with modifications of (Nordeng, 1994). Boundary layer and surface processes are based on ECHAM physics package and described in Roeckner et al. (2003). Sea surface temperature (SST) and sea ice content (SIC) is prescribed by climatological monthly averaged data from the AMIP database between 1987 to 2006. This simulation is used to evaluate differences between parametrised and superparametrised climate simulations of EMAC. In order to investigate the configuration effects of the CRM several SP-EMAC runs have been performed. In each SP-EMAC run a CRM has been embedded in each of the 8192 grid columns of the GCM. Each simulation is distinguished by its configuration of the superparametrisation. Aspects that vary along the different runs are: CRM cell orientation (OR) within a GCM grid box (alignment: north-south, east-west or full 3D), the individual size of one CRM cell (4km, 2km or 1km) and the number of CRM cells within a large-scale grid box (64, 32 or 16). Each of these three attributes characterize a SP-EMAC simulation. A list of all runs is given in table 1. Configurations that have been used in previous literature are marked and referenced appropriately.

Cloud optical properties are treated with the same submodel CLOUDOPT (Dietmüller et al., 2016) for all simulations but using different input variables for CTRL and SP-EMAC simulations. The calculation is based on liquid water content, ice water content, in-atmosphere cloud cover and cloud nuclei concentration. The latter has a fixed profile exponentially decreasing with altitude (Roeckner et al., 2003). Based upon the liquid and ice water content effective radii are computed with the assumption of treating ice as hexagonal plates (Johnson, 1993; Moss et al., 1996). All resolution-dependent cloud optical parameters (asymmetry factor and cloud inhomogeneity) are kept fixed for all simulations based on the T42 GCM resolution. Regarding the control simulation, total cloud cover, liquid and ice water content are calculated within the cloud submodel (Roeckner et al., 2006). For all SP-EMAC runs these variables are calculated as domain averaged values over all CRM grid cells within a GCM column. Thereby no subgrid-scale calculation of cloud optical properties (as well as radiative tendencies) is performed. This method has been applied intentionally to use subgrid-scale information but condense it onto the coarse GCM grid using the same subsequent submodels. A further modification concerning cloud optical properties and radiative transfer would complicate the analysis to differentiate model discrepancies between SP-EMAC and CTRL. Differences could be either due to a different cloud development within the superparameterisation or cloud radiative effects considering subgrid-scale cloud fractions.

Further information on the simulations setup, namelist settings and the newly implemented CRM options is given in the supplement.

The simulation period spans 15 months considering the first three months of the simulation as spin-up and discarding it from the analysis. Monthly averaged data has been used for the evaluation. In total 21 SP-EMAC simulations have been performed to evaluate the differences that come along when changing the configuration of the superparametrisation. It is

**Table 1.** Overview of sensitivity simulations

| # | Simulation Name | Description |
|---|---|---|
| 0 | CTRL | EMAC control simulation with parametrised convection and clouds |
| | SP-EMAC | SP-EMAC simulations with diverse configurations specified by three abbreviations: |
| 1[a,f,i] | OR1 4km 64 | |
| 2[b,d,e,g,j] | OR1 4km 32 | *abbr. #1: CRM orientation* |
| 3 | OR1 4km 16 | orientation of CRM cells within a GCM cell |
| 4 | OR1 2km 64 | OR1, OR2 or OR3 |
| 5 | OR1 2km 32 | OR1 = east-west orientation |
| 6[d,i] | OR1 1km 64 | OR2 = north-south orientation |
| 7 | OR1 1km 32 | OR3 = three dimensional (3D) CRM |
| 8[c,d] | OR2 4km 64 | |
| 9[h] | OR2 4km 16 | *abbr. #2: CRM grid size* |
| 10 | OR2 2km 64 | 4km, 2km or 1km |
| 11 | OR2 2km 32 | |
| 12 | OR2 2km 16 | *abbr. #3: number of CRM grid cells* |
| 13 | OR2 1km 64 | 64, 32 or 16 |
| 14 | OR2 1km 32 | |
| 15 | OR2 1km 16 | |
| 16[b] | OR3 4km 64 | for the 3D orientation the CRM cells are arranged as follows: |
| 17 | OR3 4km 32 | total cells = number of cells in east-west direction **x** number of cells in north-south direction |
| 18 | OR3 2km 32 | 64 cells = 8 **x** 8 |
| 19 | OR3 2km 16 | 32 cells = 8 **x** 4 |
| 20 | OR3 1km 64 | 16 cells = 4 **x** 4 |
| 21[g] | OR3 1km 16 | |

Configurations that have been used in previous literature: [a]Khairoutdinov and Randall (2001), [b]Khairoutdinov et al. (2005), [c]Khairoutdinov et al. (2008), [d]Kooperman et al. (2013), [e]Kooperman et al. (2014), [f]Cole et al. (2005), [g]Parishani et al. (2017), [h]Pritchard et al. (2014), [i]Marchand and Ackerman (2010), [j]Wang et al. (2011a)

noteworthy to mention that no tuning is done, thereby allowing the simulation to react to its own dynamics and inter-dependencies. This is done on purpose to derive the distinct consequence of a different CRM configuration. In order to condense the information of all superparametrised runs an ensemble depictive representation is used to display the mean performance (black line) as well as the variability (grey area) of all SP-EMAC simulations. Thereby, figures always show the ensemble average of all SP-EMAC runs if not mentioned otherwise.

## 3 Evaluation

The evaluation of SP-EMAC is divided in three parts. The first section covers a global analysis of SP-EMAC comparing mean global variables and their variability. Secondly, regional aspects are investigated revealing a higher importance of the CRM setup to local fields. The last part explains issues of several configurations of the superparametrisation and their impact on a global scale.

### 3.1 Global aspects

The first evaluation of the new model system covers the comparison of different mean global variables and their spatial and temporal distribution of SP-EMAC with the control simulation (CTRL) and several observations. Table 2 lists global mean values of top of the atmosphere (TOA) net radiative flux ($F_{net}$), surface temperature over land ($T_s$), total cloud cover ($C_{tot}$), precipitation ($P$), liquid water path (LWP), ice water path (IWP) and the net cloud radiative effect (NetCRE) at TOA. These variables indicate the overall performance of all SP-EMAC simulations for the first time without tuning to relevant climate measures. In order to classify these single-year averages, two additional multiyear simulations have been conducted using configuration OR1 4km 64 and OR2 1km 16. These simulations help to estimate the annual variability for SP-EMAC simulations. The associated standard deviations are given in Table 2.

Considering the radiative fluxes at TOA almost all configurations of the superparametrisation lie within a range of $\pm 4$ W/m$^2$ reflecting an almost balanced radiation budget. Only two setups (OR3 1km 64 and OR3 1km 16) show a strong negative imbalance generated by too reflective clouds. The energy deficit for these simulations can be explained by a large negative net cloud radiative effect dominated in

**Table 2.** Overview of different global mean variables (values of $T_s$, LWP and IWP represent averages between $60^\circ$ latitudes). Quantities related to radiative fluxes ($F_{net}$ and NetCRE) represent top of the atmosphere (TOA) values.

| Simulation Name | $F_{net}$ (W/m$^2$) | $T_s$ (K)* | $C_{tot}$ (%) | $P$ (mm/d) | LWP (g/m$^2$) | IWP (g/m$^2$) | NetCRE (W/m$^2$) |
|---|---|---|---|---|---|---|---|
| CTRL | 3.7 | 289.6 | 60.0 | 2.9 | 92 | 28 | -22.7 |
| OR1 4km 64 | -2.1 | 289.4 | 57.2 | 3.2 | 98 | 52 | -28.5 |
| OR1 4km 32 | -3.7 | 289.3 | 57.6 | 3.2 | 105 | 54 | -30.2 |
| OR1 4km 16 | -5.3 | 289.1 | 59.0 | 3.2 | 113 | 56 | -31.8 |
| OR1 2km 64 | 0.6 | 289.5 | 56.8 | 3.1 | 91 | 56 | -26.3 |
| OR1 2km 32 | -1.4 | 289.3 | 57.0 | 3.2 | 99 | 56 | -28.1 |
| OR1 1km 64 | 2.2 | 289.4 | 56.0 | 3.1 | 89 | 57 | -25.0 |
| OR1 1km 32 | -3.2 | 289.1 | 57.6 | 3.2 | 103 | 54 | -29.8 |
| OR2 4km 64 | 0.3 | 289.5 | 57.7 | 3.1 | 93 | 53 | -26.4 |
| OR2 4km 16 | -3.7 | 289.2 | 58.2 | 3.2 | 98 | 55 | -30.2 |
| OR2 2km 64 | 1.7 | 289.5 | 56.4 | 3.1 | 89 | 54 | -25.3 |
| OR2 2km 32 | -0.1 | 289.4 | 57.0 | 3.2 | 96 | 56 | -26.9 |
| OR2 2km 16 | -1.9 | 289.5 | 58.0 | 3.2 | 104 | 58 | -28.6 |
| OR2 1km 64 | 2.2 | 289.6 | 55.5 | 3.2 | 88 | 56 | -24.7 |
| OR2 1km 32 | 0.9 | 289.7 | 56.2 | 3.2 | 94 | 57 | -25.7 |
| OR2 1km 16 | -1.3 | 289.5 | 58.4 | 3.2 | 101 | 59 | -27.8 |
| OR3 4km 64 | -0.5 | 289.4 | 57.4 | 3.2 | 94 | 57 | -27.4 |
| OR3 4km 32 | -2.5 | 289.7 | 57.9 | 3.2 | 100 | 58 | -28.9 |
| OR3 2km 32 | -2.6 | 289.9 | 59.0 | 3.2 | 102 | 57 | -30.3 |
| OR3 2km 16 | -4.7 | 289.7 | 62.0 | 3.2 | 110 | 60 | -32.7 |
| OR3 1km 64 | -6.8 | 289.9 | 59.2 | 3.1 | 109 | 51 | -34.5 |
| OR3 1km 16 | -11.4 | 289.8 | 64.2 | 3.1 | 124 | 55 | -39.5 |
| std[+] | 0.2 / 0.3 | 0.2 / 0.2 | 0.2 / 0.4 | 0.1 / 0.1 | 0.7 / 1.1 | 0.3 / 2.5 | 0.2 / 0.3 |
| Observations | $0.8 \pm 0.4^a$ | $288.9^b$ | $62.5 \pm 4.4^c$ | $2.6 \pm 0.4^d$ | $30 \pm 10^e$ | $39 \pm 20^e$ | $-20.9 \pm 4.0^a$ |

* model average surface temperature over land is represented by values of the lowermost model layer
[+] standard deviations of two multiyear simulations each spanning 10 years. Configurations OR1 4km 64 and OR2 1km 16 have been utilized for these long runs.
[a] CERES EBAF-TOA Edition 2.8 (Clouds and Earth's Radiant Energy System - Energy Balanced and Filled) - 04/2000-03/2010, Wielicki et al. (1996); Loeb et al. (2009)
[b] NCEP/DOE2 Reanalysis data provided by the NOAA/OAR/ESRL PSD, Boulder, Colorado, USA, from their Web site at https://www.esrl.noaa.gov/psd/ - 01/1979-12/2010, Kanamitsu et al. (2002)
[c] CERES ISCCP-D2LIKE-MERGED - Edition 3A, NASA Langley Atmospheric Science Data Center DAAC. DOI: 10.5067/Aqua/CERES/ISCCP-D2LIKE-MERG00_L3.003A - 04/2000-03/2010, Wong, T. (2008)
[d] Global Precipitation Climatology Project (GPCP) Climate Data Record (CDR), Version 2.3 (Monthly). National Centers for Environmental Information. DOI:10.7289/V56971M6 - 01/1981-12/2010, Adler et al. (2018)
[e] CM SAF CLARA-A2 (The Satellite Application Facility on Climate Monitoring Cloud: Albedo And Surface Radiation dataset from AVHRR data – second edition) - 04/1986-03/2010, DOI:10.5676/EUM_SAF_CM/CLARA_AVHRR/V002, Karlsson et al. (2017)

the shortwave and an overestimation of LWP. Additionally, it should be mentioned that the high imbalance is only seen for the 3D-setups of SP-EMAC. Changing the size or number of cells in a three-dimensional CRM setup drastically changes the covered area of the superparametrisation. This modification (reduction in CRM area) seems to significantly influence the CRM properties to correctly simulate the mean effects of subgrid-scale processes within a GCM cell. A similar 3D CRM configuration (in terms of CRM domain size) is applied in Parishani et al. (2017) with the exception of using a much higher horizontal (250 m) and vertical (down to 20 m) resolution. Focusing on shallow cumulus clouds this work presents improved cloud cover profiles of boundary layer clouds but an enhanced shortwave cloud radiative effect resulting in a strong global mean bright bias. The latter result is comparable to the imbalanced simulations mentioned above. An ev-

ident cause is the restricted CRM domain size that plays a crucial role for the development of deep convection and associated high clouds by cold pools and mesoscale organization (Tompkins, 2001). Another possible feedback that could degrade global statistics, affecting large-scale dynamics for all OR3 simulations, is the momentum transport, which is different in comparison to the two dimensional CRM setups. Nevertheless, these simulations (#20 and #21 in table 1) are discarded from further analysis because the mean climate is highly deteriorated. Concerning the range of averaged surface temperature over land (neglecting Arctic and Antarctica) values between 289 and 290 K mirror the variability of the SP-EMAC ensemble. All simulations including the control simulations depict a higher surface temperature compared to reanalysis data. The difference is partly due to the model output variable that presents the temperature of the lowermost

model layer instead of using the 2m-temperature. The variability in mean surface temperature over land emphasizes the influence on the planetary boundary layer due to a change in the hydrological cycle. Previous studies (Qin et al., 2018; Sun and Pritchard, 2018) have shown that land-atmosphere coupling is improved using a superparameterisation. Nevertheless, these results rely on one specific CRM setup. The one degree variability in near surface temperature suggest a not negligible effect on the hydrological and thermal coupling due to different CRM configurations.

In contrast to $T_s$, the variability in mean global precipitation ($P$) occurs small. Almost all SP-EMAC configurations display similar global averaged precipitation amounts. All superparameterised simulations show slightly overestimated precipitation rates in comparison with GPCP (Global Precipitation Climatology Project) data and its uncertainty. In a global context the CRM configuration does not have an effect on annual mean precipitation, but significant differences occur spatially depending on the chosen setup (see section 3.2). The total cloud cover for all superparameterised simulations is underestimated by 5 % with the current setup of SP-EMAC. Similar underestimations in cloud amount and overestimation in cloud optical depth (see section 3.3) have been observed in past multiscale modeling framework (MMF) studies (Marchand and Ackerman, 2010; Parishani et al., 2017). However a cloud coverage around 57 % still lies within the range (at the lower end) of current estimates of several GCMs participating in the Coupled Model Intercomparison Project Phase 3 and Phase 5 (CMIP3 and CMIP5; Probst et al. (2012); Calisto et al. (2014)). Nevertheless, this deficit can be compensated by further tuning efforts as it has been done for the control simulation depicting a mean total cloud cover of 60 % (Mauritsen et al., 2012). Because deficiencies in cloud amounts are closely related to the liquid and ice water path even higher differences are expected to arise. Best estimates for globally averaged LWP (IWP) based on different observational data sets expose a highly uncertain range between 30-50 g/m$^2$ (25-70 g/m$^2$) with an upper limit of 100 g/m$^2$ (140 g/m$^2$) (Jiang et al., 2012). These differences are due to different satellite sensor sensitivities, attenuation limits, retrieval errors and algorithmic assumptions, therefore showing no clear consensus throughout the literature (O'Dell et al., 2008; Stubenrauch et al., 2013). Comparing AVHHR (Advanced Very High Resolution Radiometer) satellite data with model results displays high discrepancies in liquid and ice partitioning. Similar to the control run all SP-EMAC configurations show a comparable mean LWP around 90 to 110 g/m$^2$. These high amounts of liquid water in the atmosphere seem to extremely overestimate the underlying observations of CM SAF (The Satellite Application Facility on Climate Monitoring) but are on the upper range of current LWP estimates and GCM simulations (Lauer and Hamilton, 2013). Previous studies have shown relatively high amounts of LWP and IWP for SP-CAM as well (Wyant et al., 2006b; Parishani et al., 2018). The most

likely reasons of very high IWP values (and LWP values) are due to parameter settings within the CRM's cloud microphysics. Because, opposing to the control simulation the new model was not tuned to a proper extent, setting the parameters within the cloud microphysics (i.e. autoconversion thresholds of ice/liquid, terminal fall velocity or the temperature bounds of ice-liquid partitioning) could easily lead to lower (more realistic) values (see Parishani et al., 2018, Table 2). Because we have used the same parameters as noted in Table 1 (simulation SPCTRL) of Parishani et al. (2018) the IWP/LWP are on the upper range of observations. Nevertheless, tuning of several parameters could lead to unintended effects because of nonlinear interactions, thereby influencing other quantities like cloud cover or cloud radiative effect. The physical processes during model integration of rationing cloud water into its liquid and ice phase is a compensating effect on total cloud cover and radiation. In addition to LWP/IWP estimates, precipitable water (i.e. integrated water vapor) displays a range between 24.9 and 25.9 mm for all SP-EMAC runs (CTRL: 24.9 mm), whereas most simulations lie within 24.9 and 25.6 mm. These values lie within observational constraints and past model studies (Demory et al., 2014).

One major last aspect to consider is the net radiative effect of clouds that is affected by the total cloud cover as well as their optical thickness and vertical extent. Absorption and reflectance of solar and terrestrial radiation is influenced by the presence of clouds and the total net cloud radiative effect (at TOA) can be quantified as the sum of its shortwave and longwave component:

$$\mathrm{NetCRE} = \mathrm{CRE_{SW}} + \mathrm{CRE_{LW}}$$
$$\mathrm{CRE_{SW}} = (\mathrm{SW^{\downarrow}} - \mathrm{SW^{\uparrow}_{all}}) - (\mathrm{SW^{\downarrow}} - \mathrm{SW^{\uparrow}_{clear}})$$
$$= \mathrm{SW^{\uparrow}_{clear}} - \mathrm{SW^{\uparrow}_{all}}$$
$$\mathrm{CRE_{LW}} = (\mathrm{LW^{\downarrow}_{all}} - \mathrm{LW^{\uparrow}_{all}}) - (\mathrm{LW^{\downarrow}_{clear}} - \mathrm{LW^{\uparrow}_{clear}})$$
$$= \mathrm{LW^{\uparrow}_{clear}} - \mathrm{LW^{\uparrow}_{all}}$$

where $\mathrm{SW^{\uparrow}_{clear}}$ and $\mathrm{SW^{\uparrow}_{all}}$ describe the clear-sky and all-sky reflected shortwave radiation at TOA and $\mathrm{LW^{\uparrow}_{clear}}$ and $\mathrm{LW^{\uparrow}_{all}}$ represent clear-sky and all-sky outgoing longwave radiation (OLR) at TOA.

The NetCRE is negative describing an overall cooling effect of clouds on the atmosphere. Concerning all simulations CTRL shows with -22.7 W/m$^2$ a net cloud radiative effect closest to the observed value of -20.9 W/m$^2$ (cf. table 2 NetCRE observed by CERES EBAF-TOA Edition 2.8). The SP-EMAC simulations cover a range between -24.7 to -32.7 W/m$^2$ which seems even more surprising because total cloud cover is slightly reduced in all superparametrised simulations. This change would usually lead to a smaller NetCRE, which is not the case. Therefore, optical properties of clouds must have substantially been changed in all SP-EMAC runs,

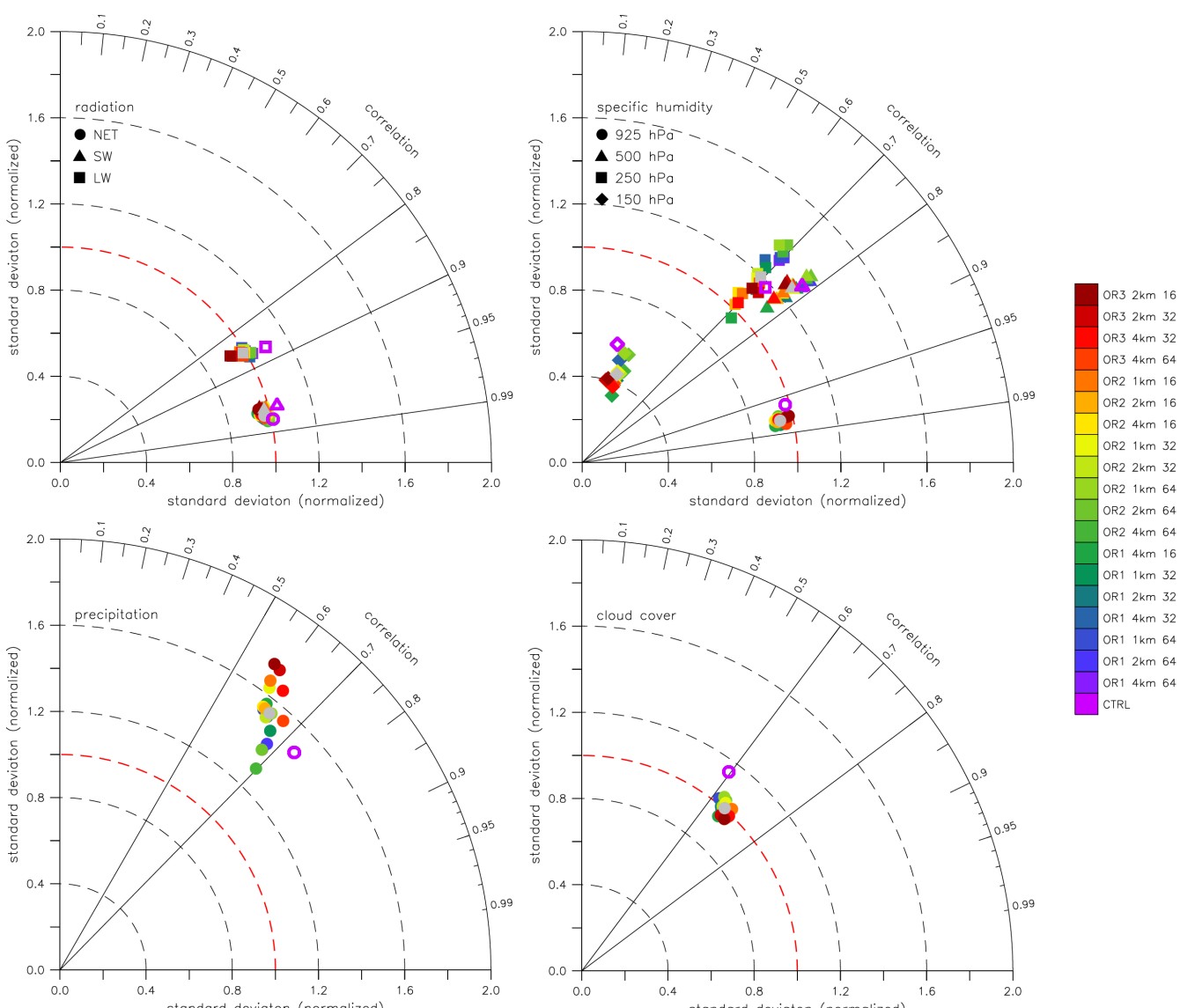

**Figure 2.** Taylor diagrams summarizing radiative fluxes at TOA, specific humidity on selected pressure levels, precipitation and total cloud cover. Individual simulations are color-coded, whereas grey markers represent the overall SP-EMAC ensemble. The control simulation is marked with open purple-white symbols. Observational data for radiation at TOA, total cloud cover and precipitation is the same as indicated in table 2. For specific humidity NCEP/DOE2 Reanalysis data is used from 01/1979 to 12/2010 and evaluated only over continental points.

indicating an increased reflection of radiation by clouds. This is evaluated in more detail in chapter 3.3. All in all, without tuning of SP-EMAC, almost all CRM configurations of SP-EMAC show mean climate characteristics equivalent to the control simulation and lie within a comparable range to observational estimates.

Apart from the analysis of averaged global fields, figure 2 displays normalized Taylor diagrams (Taylor, 2001) for four different quantities. These type of diagrams condense various aspects of multidimensional variables in comparison to observational data in one plot. The similarity of simulated and observed patterns is quantified in terms of correlation, standard deviation and centered root-mean-square error (RMSE). In total the correlation ($R$) given by the azimuthal angle, the standard deviation ($\sigma$), which is proportional to the radial distance from the origin and the RMSE corresponding to the distance from the observational point (which is aligned at a unit distance from the origin along the x-axis) quantify the degree of agreement between modeled and observed fields. The correlation coefficient include spatial as well as temporal correlation for all variables based upon monthly averages reflecting the pattern concurrence in time and space. The Taylor diagram characterizes the similarity based upon statistical relationships between modelled and observed fields. In order

to compare these fields all observations are remapped onto the applied model resolution (T42 $\approx 2.8°$ at equator).

All simulations show shortwave and net radiative fluxes at TOA, that are in close agreement to observed fields ($R \approx$ 0.98). Further focusing on the shortwave and net radiative flux, no significant improvement of SP-EMAC in comparison with CTRL can be deduced from the Taylor diagram but an overall very similar global skill is achieved. Concerning longwave TOA radiative fluxes the correlation is slightly reduced ($R \approx 0.86$) and all SP-EMAC runs demonstrate a better variance than CTRL.

A significantly improved performance in terms of correlation and variability is also visible for total cloud cover reducing the centered RMSE by 10 %. The latter is a direct result of an improved representation in northern hemispheric total cloud cover (not shown), whereas tropical and large ocean fractions show an underestimation for the superparametrised simulations. The improvements in radiation and total cloud coverage suggest a better representation of cloud-radiation processes caused by the ability to include subgrid-scale cloud dynamics.

Comparing the continental humidity distribution for four atmospheric levels, interesting features appear. In order to compare all simulations reanalysis of NCEP has been used as quasi-observations on a global scale. Lower level specific humidity at 925 hPa show a high correlation ($R \geq 0.95$) and a comparable standard deviation for many SP-EMAC runs against reanalysis data. All superparameterised model runs show a higher correlation than CTRL and are bundled together for the lowermost tropospheric level without an apparent spread. This behaviour reflects the importance of interactions between boundary layer processes and precipitating fields. An even bigger spread is visible for mid-level and upper troposphere humidity at 500 and 250 hPa. Two features are prevailing: a decrease in correlation with increasing altitude and a higher variance for almost all simulations. The overestimated variability of specific humidity is mainly a cause of too much water vapor transport over tropical continents and not enough over tropical oceans. The decrease in the correlation coefficient expresses the difficulty to simulate the appropriate water vapor transport for higher atmospheric levels, especially in the intertropical convergence zone (ITCZ). Moreover, it is obvious that different SP-EMAC configurations have a strong impact on the upper tropospheric moisture budget at 250 hPa. This is a consequence of contrasting CRM resolved strength of vertical winds. Evaluating specific humidity distribution at the tropical tropopause level near 150 hPa, almost no correlation remains and variability in these heights is strongly underestimated. This uncorrelated relationship is negatively influenced by an almost unresolved stratospheric circulation because of the sparse vertical resolution in these heights. This is an indication of almost no water vapor transport via the Brewer-Dobson circulation from the tropics to the poles.

The representation of precipitation and its spatial and temporal distribution is slightly worse compared to the CTRL simulation with correlations less than 0.7. Furthermore, the configuration of SP-EMAC strongly modifies the distribution of rainfall within the CRM columns. A much bigger spread is visible in the Taylor diagram for precipitation comparing individual SP-EMAC runs. This pinpoints the importance of the CRM configuration onto the global precipitation distribution and will be explored in more detail focusing on regional differences in the next section.

Recapitulating the evaluation of global SP-EMAC fields, several Taylor diagrams show a similar spatial and temporal correlation compared to the control simulation with parametrised convection. A slight improvement in the longwave radiative flux and total cloud cover distribution is seen in the Taylor diagrams. The representation of precipitating

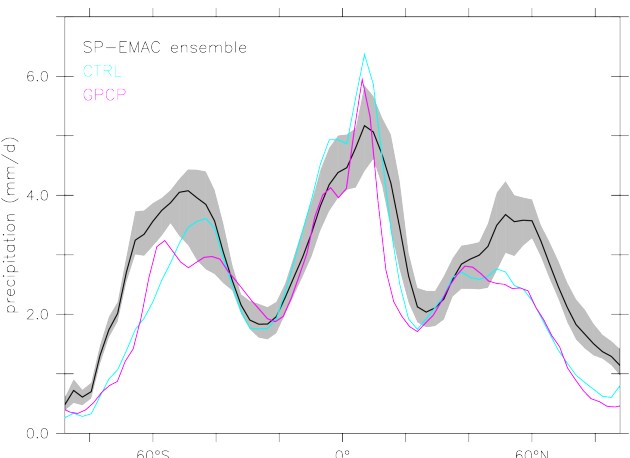

**Figure 3.** Zonal averaged simulated precipitation compared to GPCP data.

fields is slightly deteriorated. Accompanying tuning efforts for SP-EMAC could easily outperform the control simulation and thereby showing the advantage of resolving small-scales features and their impact onto global metrics.

## 3.2 Influence on regional aspects

The introduction of a superparametrisation resolving cloud dynamics in a GCM explicitly implies changes of local phenomena like precipitation, cloud regimes or boundary layer characteristics. This section evaluates regional patterns of precipitation and cloud radiative effects of SP-EMAC. In addition to that, the diurnal cycle of tropical precipitation is diagnosed as well as probability density functions (PDFs) for specified regions.

As a first step, significantly different precipitating regions for all simulations are identified and compared to observations. Moreover facing current deficiencies of GCMs, two specified regions are taken into account to analyse simulated precipitation features: maritime tropics (in particular the Warm Pool region) and the southern mid-latitudes. In previous literature

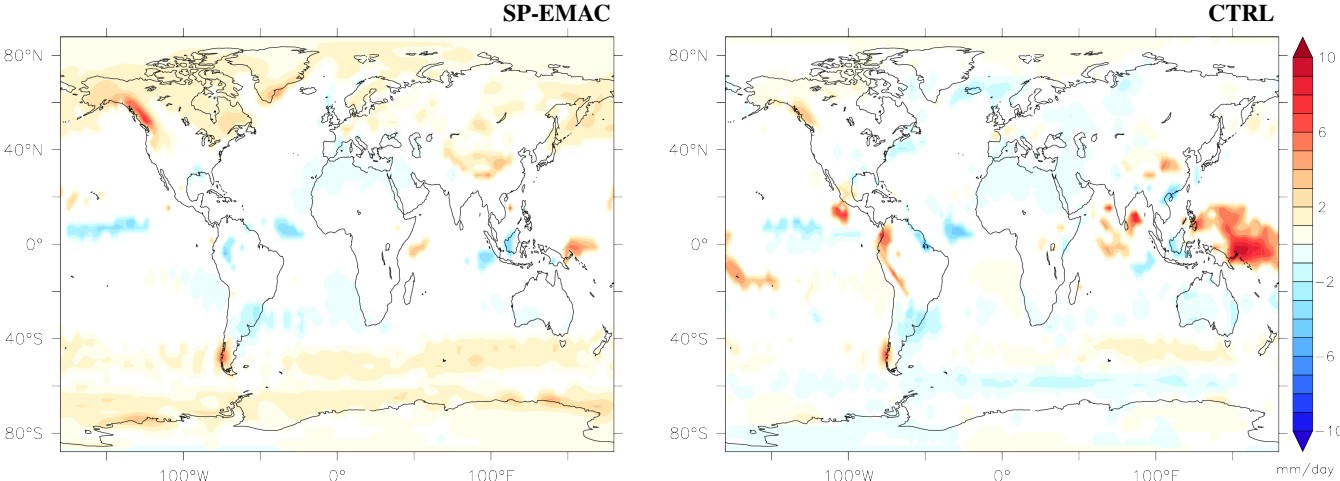

**Figure 4.** Differences in precipitation of SP-EMAC ensemble (left panel) and CTRL (right panel) in contrast to global observations (GPCP). Colored areas show only regions with significant differences in precipitation (analysed with t-Tests on a significance level of 90%).

it has been shown that the maritime continent depicts too much precipitation for all CMIP5 models consistently (Flato et al., 2013). In addition to that, simulating the Asian summer monsoon tends to overestimate rainfall amounts by a too strong convection-wind-evaporation feedback using a convection parameterisation or a superparameterisation (Luo and Stephens, 2006). Complementary, an overestimation in oceanic precipitation frequency is simulated over the southern hemisphere indicating too much drizzle (Stephens et al., 2010). Although a new study suggested that these biases originate from processes other than convection, a reduction of these errors is clearly accomplished by using convection parametrisations (Maher et al., 2018). The comparison of SP-EMAC with observations and a parametrised control simulation will reveal the importance of resolving subgrid-scale dynamics in a superparametrised GCM for these regional improvements.

Figure 3 shows zonal averaged precipitation rates for SP-EMAC, CTRL and GPCP data. In correspondence figure 4 highlights regions with significant differences in annual mean precipitation compared to observations. These regions have been identified by a couple of t-Tests on a significance level of 90%. For the control simulation one single t-Test has been carried out to emphasize important areas. Considering the analysis for SP-EMAC, regions are highlighted when more than half of all superparametrised simulations show a significant difference between observed and modelled fields. The control simulation is in close agreement to the GPCP observations with the exception of enhanced tropical precipitation, which is well represented by the superparametrisation. Contrary to this, an overestimation in the northern and southern mid-latitudes is visible for SP-EMAC, independent of the chosen CRM configuration. This finding is in agreement with the study of Marchand et al. (2009), showing an overestimation of low-level

hydrometeors in mid-latitude storm tracks using the same superparametrisation within SP-CAM (Superparametrised - Community Atmosphere Model). An improvement is given by Kooperman et al. (2016), showing no systematic biases within the mid-latitudes using a two-moment microphysical scheme linked to aerosol processes (Wang et al., 2011a). Regardless of these studies, SP-EMAC sensitivity runs suggest that formation of precipitation including the ice phase (or mixed-phase) is substantially better simulated than rainfall in almost pure liquid clouds, what is often the case for maritime precipitation in the southern mid-latitudes (Matsui et al., 2016). Nevertheless, a high sensitivity in precipitation is simulated within the ITCZ and the northern and southern mid-latitudes depending on the CRM configuration. Analysing this in more detail, it emerges that the contribution of this variability is mostly generated above the oceans and coastal regions (not shown). That implies that simulated precipitation rates are sensitive to land-ocean contrasts.

Focusing on two specific regions, figure 5 displays probability density functions of monthly precipitation rates for the Warm Pool Region and the southern ocean mid-latitudes. The former is defined as an area where sea surface temperatures exceed 297 K and strong convective systems develop, whereas the latter defines only oceanic regions in the southern hemisphere between 36° and 64°, characteristically associated with marine boundary layer clouds (Mace, 2010; Haynes et al., 2011). Embedded maps present the spatial distribution for the chosen region as yearly averaged precipitation rates. In addition to that, the non-precipitating fraction (below 1 mm/d) is shown as bars including the variability of SP-EMAC, induced when choosing different configurations and the interannual variability of the observational data. Based on the overall improvement for SP-EMAC to simulate precipitation in the Warm Pool region the PDF shows

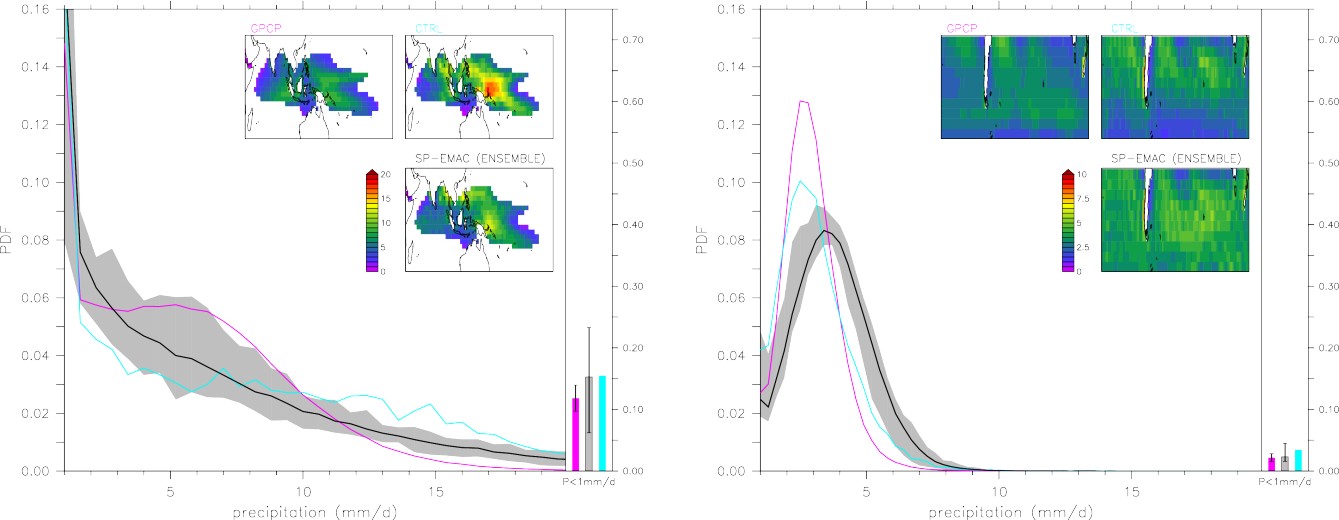

**Figure 5.** PDFs of monthly precipitation rates for the Warm Pool region and southern ocean mid-latitudes comparing to 30-years of GPCP data (same as in table 2). Inserted maps show yearly averaged rates in mm/d for the specified geographical regions as pointed out in the text. Line colors correspond to the colors above the inserted panels. On the right hand side of each figure the fraction of non-precipitating cells (below 1 mm/d) is displayed, including the interannual variability for the GPCP data and the variability induced by different SP-EMAC configurations (uncertainty bars). Note: x-axis begins at 1 mm/d.

important characteristics, that are to some extent reproduced by the superparametrisation. The most distinct feature for the maritime continent PDF is the high variability for the SP-EMAC simulations covering almost the entire range of observed probabilities. Compared to the control simulation it is most obvious that high precipitation rates (above 10 mm/d) are better represented by the CRM. Precipitation rates between 3 and 10 mm/d are underestimated by almost all simulations. Depending on the chosen configuration it can be concluded that the configurations OR1 4km 64, OR1 2km 64, OR2 4km 64 and OR2 2km 64 produce the best estimate of precipitation in the Warm Pool region (not shown). Each of these simulations show enhanced precipitation probabilities between 5 to 10 mm/d and produce the lowest probabilities for high precipitation rates in agreement with the GPCP observations (not shown). Comparing the spatial distributions a single maximum precipitation spot is visible in the western pacific when using the convection parametrisation. This is not as prominent for the SP-EMAC simulations displaying a more widespread distribution. At last, non-precipitation probabilities (comparing boxes at the right side of the plot) are in close agreement with the GPCP data but expose a huge variability for SP-EMAC reflecting the strong dependence on the chosen configuration.

The comparison of precipitation rates in the southern hemisphere mid-latitudes reveals two systematic problems of SP-EMAC: an underestimation of lower precipitation rates (between 1 to 5 mm/d) and a shift in peak precipitation rate from the observed value of 2.5 mm/d to almost 4 mm/d, explaining the overestimation in figure 3. This feature is significant for all superparametrised simulations indepen-

dent of the chosen CRM configuration. Furthermore, the comparison of almost non-precipitating grid cells reveals a similar amount of dry days in comparison with the control simulation. This finding is in agreement with other models showing a similar behaviour between parametrised and superparametrised simulations (see Kooperman et al., 2016, supplement S2). All in all, the control simulation can reproduce the peak precipitation, whereas it is skewed to larger values (above 4 mm/d). Pointing out the differences in the microphysics one has to consider the different auto-conversion rates used within the CRM cells and within the cloud scheme of the control run. The superparametrisation uses a simple fixed conversion rate (see Khairoutdinov and Randall, 2003, Appendix D), whereas the cloud scheme uses the formulation of Beheng (1994). Focusing on this aspect, Suzuki et al. (2015) has shown that the distribution of precipitation categories (non-precipitating, drizzle, rain) is dependent on its expression, thereby influencing the precipitation rate. Future studies with SP-EMAC should investigate the onset of precipitation for maritime clouds in more detail or should consider using a two-moment microphysical scheme and its coupling to an aerosol submodel.

Apart from the distribution of precipitation, a known problem of GCMs is the incorrect representation of the diurnal cycle in precipitation within the tropics (Collier and Bowman, 2004; Guichard et al., 2004; Dai, 2006). Improvements have been suggested by Bechtold et al. (2004) for convection parametrisations based on their entrainment rates. Additionally, superparametrised GCMs have been studied and show progress in representing the diurnal cycle of precipitation and its contrast between ocean

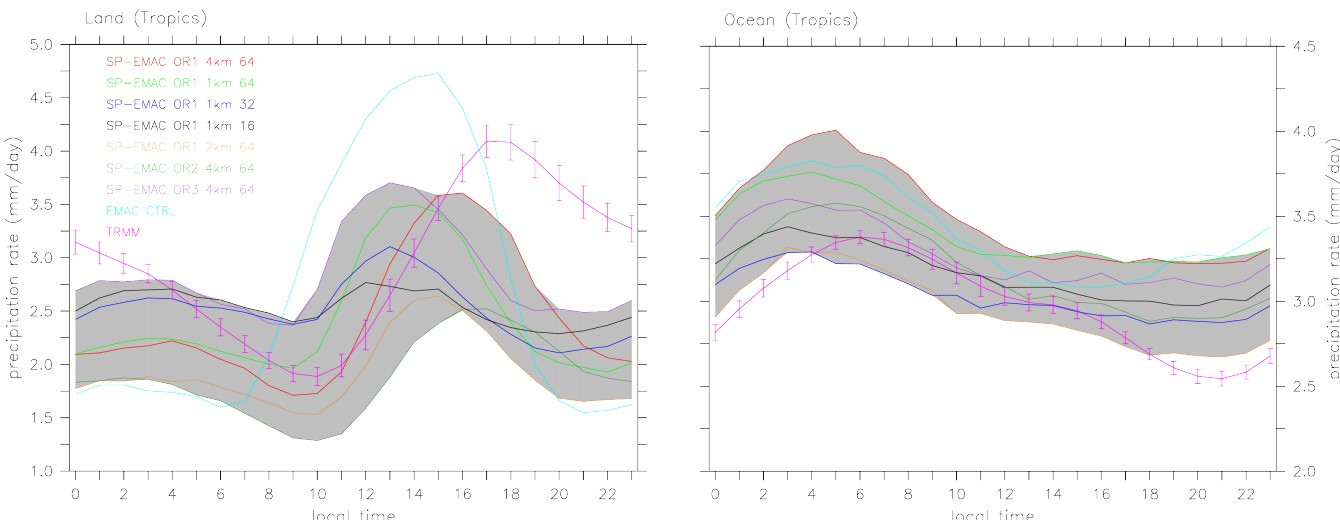

**Figure 6.** Comparison of diurnal tropical precipitation (30° around the equator) for land and ocean with 3-hourly TRMM data* between 1998 and 2010 for the month July. The observational standard deviation is shown by error bars indicating interannual variability in the diurnal cycle. The grey area covers the variability of all SP-EMAC configurations.

* Tropical Rainfall Measuring Mission (TRMM) (2011), TRMM (TMPA) Rainfall Estimate, L3, 3 hour, 0.25°x0.25°, V7, Greenbelt, MD, Goddard Earth Sciences Data and Information Services Center (GES DISC), Accessed: 02/11/2018, DOI: http://doi.org/10.5067/TRMM/TMPA/3H/7, Huffman et al. (2007)

and land (Khairoutdinov et al., 2005; Zhang et al., 2008; Pritchard and Somerville, 2009a, b; Tao et al., 2009). In order to analyse this process the output has been increased to produce precipitation rates on a hourly basis for one entire month (July). Instead of using the full SP-EMAC ensemble only a subset of superparametrised simulations with an annual precipitation below 3 mm/d has been chosen. These simulations have been selected because they have the smallest difference in comparison to observational data (compare with table 2). The hourly output has been compared to multi-monthly July averages of TRMM_3B42 v7 data between 1998 and 2010. Figure 6 displays the averaged diurnal precipitation transformed to local solar time (LT) for continental and oceanic grid points between 30° latitudes around the equator. Investigating the diurnal precipitation over land, observational evidence exposes a peak around 17 LT and an onset in precipitation around 9 LT. Previous studies of different TRMM products (3B42 and 3G68) have revealed a time lag, which may be related to infrared precipitation estimates (Kikuchi and Wang, 2008). It had been stated that the maximum peak in TRMM_3G68 is more reliable, occurring at 15 LT. Therefore, TRMM product 3B42 shows a lag of approximately 3 hours opposed to 3G68, which need to be taken into account for this study (Sato et al., 2009). This time lag seems to be valid for the maximum peak but not the onset in precipitation (Sato et al., 2009). The control simulation does not reproduce any of these timings, confirming the difficulty of GCMs including convection parametrisations to correctly simulate the diurnal cycle. The onset and peak of precipitation is around 4 hours

too early and the amplitude is overestimated. Many aspects of this evolution can be attributed to diminishing CIN (convecitve inhibition) during sunrise and increasing CAPE (convective available potential energy) during the day, that are the basis of triggering and sustaining the convection parametrisation. The shift of a too early precipitation onset is substantially improved using any kind of SP-EMAC simulation. Independent of the CRM configuration the timing of the onset of precipitation is almost perfectly reflected in comparison to 12-year averaged TRMM data for the month of July (see figure 6). This indicates that cloud evolution is not only coupled to the diurnal solar insolation but follows PBL evolution. In contrast, diurnal peak precipitation is completely dependent on the CRM configuration for SP-EMAC indicating values between 2.5 to 3.75 mm/d and peak time spreading from 12 LT (OR1 1km 16) to 17 LT (OR2 4km 64). Taking into account that the observational product has a lag of 3 hours, some SP-EMAC configuration provides good estimates in the timing of maximum precipitation. Furthermore, the decline in precipitation after peaking is too strong, resulting in a secondary maximum during the night (between 2 to 5 LT). This secondary peak is partly visible for the TRMM data but only for spring and autumn seasons (Yang and Smith, 2006). Even if the diurnal cycle is not captured very well, it has almost no influence on the global mean precipitation rate. One significant highlight corresponds to the different diurnal amplitudes, which increase with increasing number of CRM cells, whereas single simulations with 32 or 16 cells exhibit a small or almost no diurnal cycle in precipitation.

The diurnal cycle over tropical oceans is displayed on the right side in figure 6. The observed diurnal cycle presents a peak in precipitation around 6 LT and a clear minimum in the evening hours (21 LT). A saddle point (secondary maximum) can be identified around 14 LT. The primary mechanisms to explain this cycle are: „static radiation-convection" and „dynamic radiation-convection", that describe the process of radiative cooling while increasing thermodynamic instability enhancing nighttime precipitation or suppressing daytime rainfall through decreased convergence into the convective region. A more detailed description and further mechanisms are given in Yang and Smith (2006).

The control simulation shows an overall overestimation of oceanic precipitation rates by 0.5 mm/d but a similar timing in peak precipitation. The simulated decline in oceanic rainfall is too steep resulting in too early minimum precipitation rate around 15 LT and an increase directly afterwards. In contrast, nearly all SP-EMAC runs simulate a consistent decline as CTRL after peaking precipitation between 4 and 6 LT but remain almost constant until 21 LT. The spread in oceanic precipitation rates of the SP-EMAC ensemble is slightly lower (0.75 mm/d) compared to the diurnal cycle over land (1.25 mm/d). Analogous to the diurnal cycle over land it emerges that the amplitude of precipitation rates increases with an increasing number of CRM cells. Especially two specific configurations (OR1 2km 64 and OR2 4mk 64) are in very good agreement with TRMM data. Nevertheless, all simulations miss the representation of a secondary maximum around 14 LT. This effect could be due to neglected diurnal variations in prescribed SSTs thereby restraining ocean surface heating (Sui et al., 1997, 1998). Further investigations of 2D cloud-resolving model simulations with diurnally varying SSTs exhibit an increase in afternoon rain rates suggesting influences of ocean heating in atmospheric moistening and drying throughout the day (Cui and Li, 2009).

To complete the regional analysis of SP-EMAC, cloud radiative effects at the top of the atmosphere are investigated. Ten years of satellite data of CERES are used for comparison (see table 2). CRE is divided into its longwave and shortwave component to distinguish different radiative effects. Zonal averages of cloud radiative effects are shown in figure 7. Instead of displaying all individual SP-EMAC simulations the ensemble mean (black lines) and spread (grey area) is shown. The distribution of NetCRE shows high discrepancies in the mid-latitudes. The primary cause of this difference is induced by the shortwave component of the CRE revealing the representation of too reflective clouds in all simulations for this region. Moreover, seasonal variation for $CRE_{SW}$ are enhanced in all SP-EMAC simulations as well as in the control run. Smaller deviations for NetCRE are visible in the ITCZ for SP-EMAC compensated by even larger differences for the individual components $CRE_{SW}$ and $CRE_{LW}$. Comparing CTRL and SP-EMAC in a comprehensive sense, it emerges that overall CTRL represents the zonal

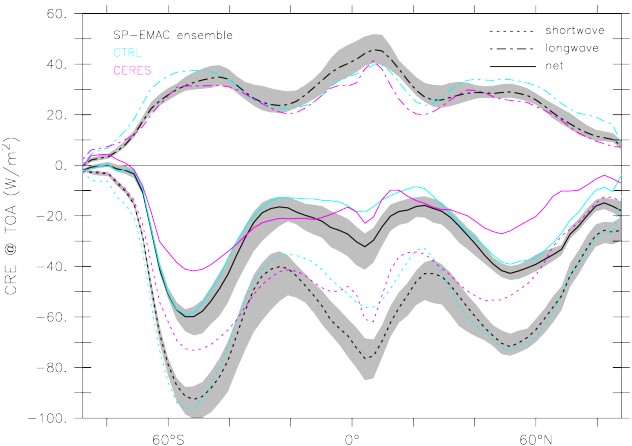

**Figure 7.** Zonal averaged simulated top-of-atmosphere cloud-radiative effects (CRE) compared to CERES EBAF-TOA data (2000-2010).

NetCRE distribution slightly better especially in the Tropics. This is sometimes due to compensating errors. On a closer look many SP-EMAC configurations improve the longwave and shortwave component in the mid-latitudes. Nevertheless, dependent on the CRM configuration, there exists high differences even in the zonal mean distribution. To identify regions with significant differences figure 8 shows absolute differences of NetCRE, $CRE_{SW}$ and $CRE_{LW}$. Similar t-Tests as for figure 4 have been performed to obtain important areas that deviate from CERES observations. Thereby, non-significant differences are shown in white. Blue areas indicate regions where cloud radiative effects are stronger (higher cooling), whereas red areas specify less cooling or even a warming effect of clouds. Comparing the differences in NetCRE maps in figure 8 it is apparent that CTRL shows larger areas of significant differences especially a positive bias over the oceans. The underestimation of cloud radiative effects for CTRL over the oceans is because of a much higher shortwave component in these regions, marking a reduced amount in low cloud cover or less reflective clouds in the areas of stratocumulus decks. Using a superparametrisation in EMAC results in smaller discrepancies for all CRE components. In particular the formerly mentioned regions demonstrate a better representation of radiative effects of low clouds. However, a significant reduction of about 10 $W/m^2$ to 20 $W/m^2$ is visible for the Western Pacific region in many SP-EMAC simulations. Evaluating the longwave and shortwave component in these region it became apparent that deep convective clouds have an increased optical depth for many CRM configurations and concurrently a higher liquid and ice water content than the control simulation (not shown). Further important differences are visible for $CRE_{LW}$ in the control simulation. A stronger warming over the complete southern ocean and the arctic region is apparent as a result of a too high liquid water path in these regions.

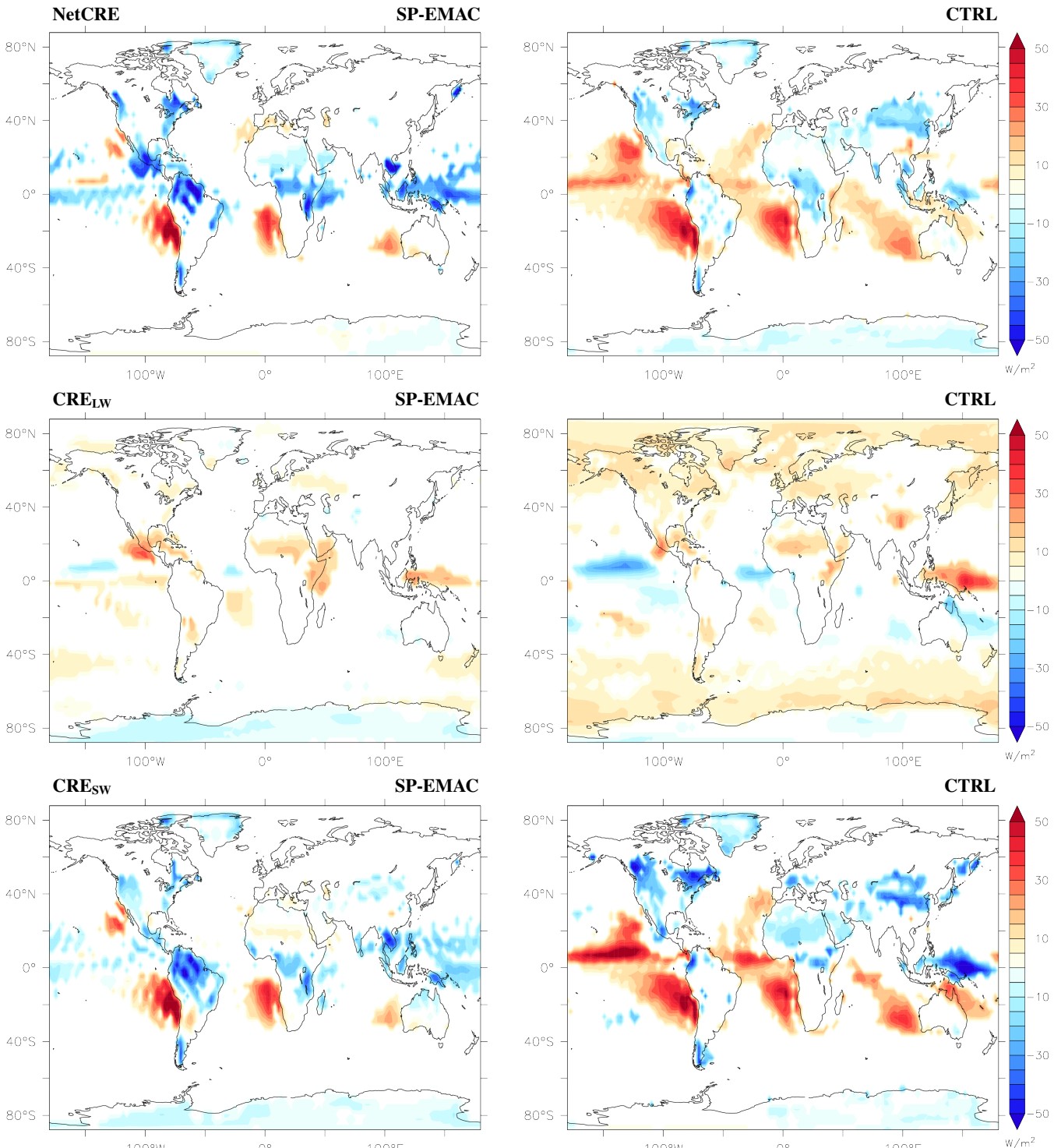

**Figure 8.** Differences in simulated minus observed cloud radiative effects at TOA (net - upper row, longwave - middle row, shortwave - lower row). Results for SP-EMAC ensemble (left column) and CTRL (right column) are shown comparing to global observations (CERES). Colored areas show only regions with significant differences in cloud radiative effects (analysed with t-Tests on a significance level of 90%).

Lastly, comparing all maps from top to bottom in figure 8, it is possible to easily identify regions that show almost no significant difference in NetCRE because of compensating errors in the longwave and shortwave part. Affected areas are: Central Africa, Central America and the Caribbean Sea for SP-EMAC and the Sahara, Greenland, North America,

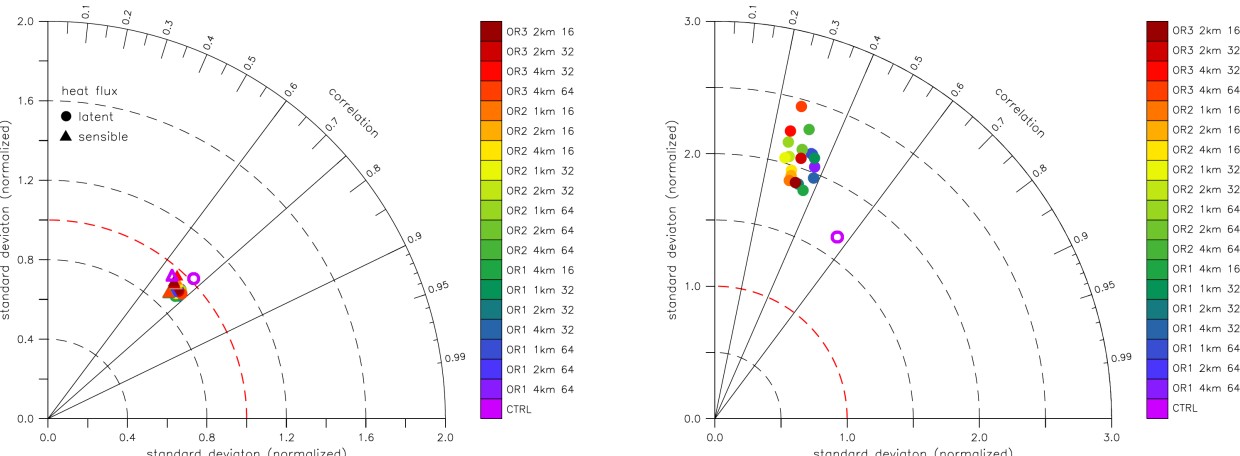

**Figure 9.** Global Taylor diagrams of sensible and latent heat fluxes (left) and cloud optical thickness (right) over land points only. The control simulation is marked with open purple-white symbols. For comparison purposes reanalysis data of NCEP is taken to compare all simulations with respect to heat fluxes and MODIS retrievals provide estimates of cloud optical properties.

the Arctic and the Southern Ocean for CTRL. Thereby, it is evident that NetCRE should not be used as a single metric to evaluate cloud radiative effects and the performance of a GCM. All in all, CRE is influenced by multiple factors: insolation, cloud amount, cloud type and surface properties (albedo). Only cloud amount and cloud type changes are relevant for explaining differences between SP-EMAC and CTRL (excluding glaciers and snow-covered areas that increase surface albedo). Even if SP-EMAC seems to reduce CRE errors, different configurations show significant different results. Future studies with SP-EMAC should always look at the different cloud radiative effects to avoid misinterpretations of model results. This is necessary because not all SP-EMAC configurations are equally appropriate to use. This addresses the need for a tuning activity for SP-EMAC in the near future.

### 3.3 Issues due to CRM's configuration

The global evaluation of SP-EMAC in chapter 3.1 has revealed some major influences of the CRM configuration onto the mean climate state. This is in particular very important concerning regional aspects of clouds and precipitation. Based upon the global mean Taylor diagrams in figure 2 no clear separation of several SP-EMAC simulations is noticeable. Tracking down other differences than mean surface temperatures and precipitation rates among SP-EMAC simulations, the distribution of surface heat fluxes is analysed. Figure 9 shows the Taylor diagram of sensible and latent heat flux over land. The fluxes are compared to NCEP reanalysis data (compare with table 2) between 1979 and 2010. Evaluating the Taylor diagram it is clearly visible that heat fluxes are well reproduced, and all simulations are bulked together. This clustering suggests that boundary fluxes are

hardly affected, even if the variability in mean global lowermost model temperature shows a not negligible range of 1 K. The analysis of heat fluxes clearly shows that the configuration (changing the size, number and orientation of the CRM) does not impact lower tropospheric properties. This independence of boundary layer fluxes on the SP-EMAC configuration is an addition to the analysis of Sun and Pritchard (2018) and Qin et al. (2018) providing a better or equal representation of the thermal and hydrologic coupling. One further modification to expand this framework could be the consideration of the large-scale directional wind speed within the CRM. All two-dimensional setups use either the zonal (OR1 cases) or meridional wind component (OR2 cases). Allowing the 2D-CRM to vary with time (Grabowski, 2004a; Cheng and Xu, 2014; Jung and Arakawa, 2016) could have an impact on the boundary layer turbulence and hence modifying the boundary layer fluxes.

Instead of analysing indirect effects like surface fluxes, global mean precipitation shows a more widely dispersed result in figure 2. In order to look at rather explicit effects of the CRM on cloud properties, it is therefore straightforward to observe cloud related variables. Therefore, cloud optical thickness (COT) for continental clouds is examined using satellite data from the Moderate Resolution Imaging Spectroradiometer (MODIS) collection 5.1. Observations between 2003 and 2015 from the combined Terra and Aqua satellites are remapped onto the coarse GCM grid and used for comparison purposes. Figure 9 shows the Taylor diagram of cloud optical thickness. As opposed to the Taylor diagram of heat fluxes a more widespread depiction of the SP-EMAC ensemble is apparent for COT. It clearly shows that cloud optical properties significantly changed within SP-EMAC in comparison to the control simulation. Generally speaking, all SP-EMAC simulations display a similar distribution as CTRL in the mid-latitudes whereas tropical cloud thick-

nesses over land are overestimated (not shown). This increase in COT partly explains the stronger cloud radiative effects for all SP-EMAC runs, compensating the overall reduced simulated total cloud cover. However, the comparison with MODIS data shows a strongly reduced correlation. A complete fair comparison with MODIS would imply the usage of the CFMIP Observational Simulator Package (COSP, (Bodas-Salcedo et al., 2011; Swales et al., 2018)). In addition to that, the neglection of subgrid-scale cloud variability is an important aspect to consider using the simulator (Song et al., 2018). Therefore, this comparison should be treated as a proxy to display the robust differences between SP-EMAC and CTRL including an observational reference for COT.

Although the representation of cloud optical thickness shows a larger spread within the SP-EMAC ensemble, it is not straightforward to identify CRM characteristics regulating the behaviour of the model. For this purpose sub-ensembles are constructed to represent distinguishable results based upon the sub-ensemble typical feature. All 19 SP-EMAC simulations are described by three configurational aspects (differentiated by three different states; see Table 1). These aspects are used to create nine sub-ensembles consisting of the same state of the configurational assignment, i.e.:

- OR1 consists of all simulations including the east-west orientation (simulation no. 1 to 7 in table 1)

- OR2 includes sim. no. 8 to 15 (i.e. north-south oriented CRM)

- OR3 covers sim. no. 16 to 19 (only full 3D CRM setups)

- 1km represents all simulations with a CRM grid size of 1 km (sim. no.: 6, 7, 13, 14 and 15)

- 2km (sim. no.: 4, 5, 10, 11, 12, 18 and 19)

- 4km (sim. no.: 1, 2, 3, 8, 9, 16 and 17)

- n16 describes all simulations with 16 CRM cells (sim. no.: 3, 9, 12, 15 and 19)

- n32 (sim. no.: 2, 5, 7, 11, 14, 17 and 18)

- n64 (sim. no.: 1, 4, 6, 8, 10, 13 and 16)

The aforementioned sub-ensembles are compared to the full SP-EMAC ensemble in order to distinguish effects that result from a specific configuration. Because cloud related properties experience larger impacts due to a different configuration full atmosphere cloud amounts are examined in Figure 10. Annual-averages of zonal-mean cloud amounts for CTRL and the full SP-EMAC ensemble are displayed in the top row as a reference. SP-EMAC simulates lower cloud top heights than the control simulation and more pronounced convective cloud coverage within the ITCZ. An increased coverage of boundary layer clouds in the southern mid-latitudes is visible. Despite the fact that major

differences between the control simulation and SP-EMAC occur, sub-ensemble are analysed and compared to the full SP-EMAC ensemble. Absolute differences are displayed in Figure 10 highlighting higher cloud amounts than the reference in red colors. The partitioning of all SP-EMAC simulations into separate sub-ensembles lead to an easy identification of significant cloud amount changes due to different CRM configurations. Sub-ensembles of OR2, 2km, 4km and n32 show only minor differences in comparison to the full SP-EMAC ensemble. Regarding the orientation cloud amounts show a slight increase in the mid-latitudes for the zonal (OR1) CRM orientation, whereas the full three-dimensional orientation imprints a significant decrease for cloud amounts above 900 hPa. Additionally, an increase of boundary layer clouds in the southern hemisphere mid-latitudes of OR3 exposes an effect of the lower tropospheric winds on cloud amount. Regarding the CRM grid size it is clearly evident that a higher CRM resolution increases the amount of boundary layer clouds (see 1km sub-ensemble in Fig. 10), which confirms the sensitivity study of Marchand and Ackerman (2010). Furthermore, a significant decrease in cloud amount is simulated for a smaller number of CRM cells (see sub-ensemble n16). Including the overall underestimation of total cloud cover for all SP-EMAC simulations, it appears that a minimum amount of 32 CRM cells is needed to provide a correct representation of cloud development within a GCM grid cell.

Similarities of sub-ensembles OR1 and n64 as well as OR3 and n16 are visible. These coincidences are due to individual SP-EMAC simulations, which demonstrate prominent features in cloud amount changes. For example an increase in boundary layer clouds is seen for OR1 4km 64, OR1 2km 64 and OR1 1km 64 (not shown), which are within both sub-ensembles of OR1 and n64. Likewise, OR3 2km 16 and OR3 1km 16 show a decrease in boundary layer clouds, resulting in a similar appearance for sub-ensemble OR3 and n16.

Overall, it should be noted that the number of ensemble members has an effect on the results because, the smaller the size of the sub-ensemble the more likely it deviates from the full SP-EMAC ensemble. Another point to mention is the vertical resolution, that has been fixed to 29 levels within the CRM co-located with the lowermost GCM levels starting at the surface. Regarding shallow cumulus clouds this resolution is too coarse to explicitly represent boundary layer cloud evolution, leading to a decrease in cloud top height and prohibiting the existence of cumulus-under-stratocumulus decoupled boundary layers (Parishani et al., 2017). Furthermore, the vertical resolution has an impact on the low cloud feedback under a warmer climate (Wyant et al., 2009; Blossey et al., 2009). Apart from low level clouds it is noteworthy that vertical resolution plays an important role for mid-level and cirrus clouds. Although superparameterised simulations improve these cloud characteristics (Wyant et al., 2006b, a) it should be kept in mind

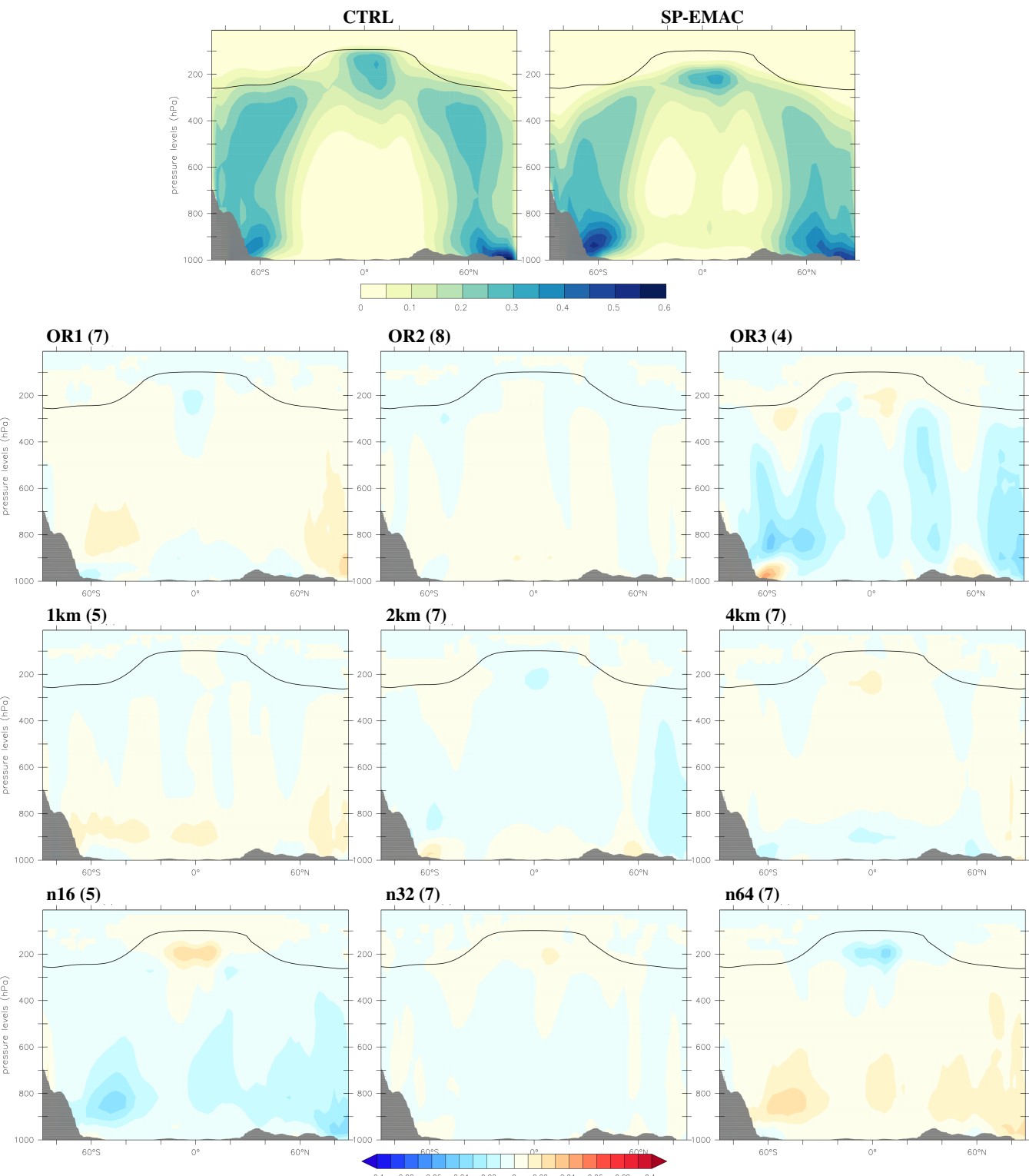

**Figure 10.** Top row: averaged zonal cloud amount for the SP-EMAC ensemble and the control simulation. The black line marks the tropopause height. Bottom rows display cloud amount differences of several SP-EMAC sub-ensembles in comparison to the SP-EMAC ensemble (top-right panel). The sub-ensembles are characterized by a CRM configuration feature, i.e.: OR1 consists of all simulations including the east-west orientation (the number of ensemble members is given in parenthesis); OR2: sub-ensemble for north-south oriented CRM cells; OR3: sub-ensemble of full 3D CRM configurations; 1km/2km/4km: SP-EMAC simulations with CRM grid size of 1, 2 or 4 km; n16/n32/n64: SP-EMAC simulations including a total number of 16, 32, 64 CRM cells.

that most cirrus clouds are diagnosed, but not explicitly represented, with the applied vertical model resolution.

In summary, it has been shown that different configurations of the CRM within SP-EMAC lead to distinctive atmospheric properties demonstrated by diverse cloud optical depths and full atmosphere cloud amount. These results suggest that cloud evolution and processing within the superparametrisation is influenced because of different CRM domain compositions. Indirect effects like the influence on global surface heat fluxes play a minor role. Even if all members of SP-EMAC show a similar performance than CTRL in terms of climate metrics evaluated in section 3.1, it should be noted that further tuning is necessary. In particular it is necessary to adjust cloud amounts and cloud optical properties. This would further improve the simulation of cloud-radiative effects and reduce the compensation of contrarily effects.

## 4    Conclusions and Discussion

The concept of embedding a cloud-resolving model into a GCM has been studied for over a decade and this paper introduces another climate model incorporating this idea. The superparametrisation based upon the System for Atmospheric Modeling (SAM, Khairoutdinov et al. (2008)) has been included into the EMAC model. This study focused on the effect of different model configurations of the embedded CRM (orientation, cell size, number of cells) on climate relevant variables. For the first time, the influence of different aspects of the superparametrisation has been systematically evaluated in 21 model simulations each spanning one year. The model runs have been compared to observations and a control simulation using a conventional convection parametrisation and a large-scale cloud microphysics scheme.

The analysis of global mean statistics for all superparametrised runs, encompassing the net radiative flux at TOA, surface temperature, total cloud cover, precipitation, LWP, IWP and the net cloud radiative effect, show similar results compared to the control simulation. Almost all global mean results lie within the range of CMIP5 models, independent of the chosen CRM configuration. Only two superparametrised setups covering a relatively small area within the GCM grid box and a three dimensional CRM orientation simulate a high energy imbalance. This supports the assumption that a minimum number of CRM grid boxes is necessary to represent cloud development covering all important subgrid-scales of a GCM. Taylor diagrams reveal improved representations of the longwave radiative flux at TOA, total cloud cover distribution and a similar distribution of atmospheric humidity using a superparametrisation in EMAC (Tulich, 2015; Tao et al., 2009). The global distribution of precipitation rates shows a degradation when comparing to GPCP data because of a too high oceanic rainfall but a better

performance for the Warm Pool region. Interestingly, a rather high influence depending on the selected CRM configuration is evident concerning precipitating fields especially over the western Pacific. Related to the analysis of rainfall PDFs the amount of non-precipitating grid cells (below 1 mm/d) is highly variable indicating contrasting onsets in precipitation for different CRM configurations for the Warm Pool and southern hemisphere mid-latitude region.

Furthermore, the diurnal cycle for tropical land and oceans has been observed separately. Independent of the configuration of the superparametrisation the onset of tropical precipitation over land is in perfect agreement with TRMM data as contrasted with the control simulation (Khairoutdinov et al., 2005; Pritchard and Somerville, 2009b). Nevertheless, the configuration of the CRM drastically changes the amplitude and peak in precipitation in the tropics. Thereby, some model setups of the superparametrisation show similar precipitation peaks in the diurnal cycle as compared to the control simulation, using a convection parametrisation with even diminishing amplitudes. This conclusion stands in contrast with recent literature proclaiming a great improvement in the simulation of the diurnal cycle using any kind of superparametrisation (Zhang et al., 2008). A rather significant feature throughout the simulations is the decreasing diurnal amplitude, when defining smaller sets of CRM cells for the superparametrised setup.

Regarding the cloud-radiation interaction it appears that the control simulation shows a slightly better representation of the net cloud radiative effect comparing the zonal distribution (Khairoutdinov et al., 2008). However, a regional analysis demonstrates that larger areas display significant differences in CRE contrasting the control simulation with the superparametrised runs. In comparison with CERES satellite data and the distribution of the longwave and shortwave CRE, it is evident that many regions show opposing effects resulting in compensating errors in the NetCRE. Many setups of the superparametrisation show improvements especially over oceanic regions for $CRE_{LW}$ and $CRE_{SW}$, but this cannot be stated for any kind of CRM setup. Further evaluation of radiative fluxes over the Southern Ocean with SP-EMAC should keep in mind the rather simplified microphysics within the CRM (Khairoutdinov et al., 2008). The partitioning of cloud ice and cloud water within a one-moment microphysical scheme cannot handle the representation of supercooled liquid clouds. These seem to have a significant effect on the solar radiation budget in this region (Bodas-Salcedo et al., 2016). The option to switch the microphysical scheme to a two-moment scheme has been added in a newer version of SP-EMAC and provides new possibilities to study these effects.

A major consideration in this study has been the issues associated with changes in CRM orientation, size or the numbers of small-scale grid cells as proposed by several other studies (Tulich, 2015; Pritchard et al., 2014; Tao et al., 2009). It has been shown that global lower tropospheric features, in partic-

ular surface heat fluxes, are hardly influenced when changing CRM aspects. These results support the research of Sun and Pritchard (2018) and Qin et al. (2018) showing an improvement in thermal and hydrologic coupling using a superparameterisation in one explicit configuration. Opposed to these indirect effects of the CRM onto climate relevant variables, cloud related properties display a significant spread due to different CRM configurations. Evaluation of full atmosphere cloud amounts suggests that a minimum number of 32 CRM cells is required to improve and account for a realistic cloud development within a GCM cell (as hypothesized by Pritchard et al. (2014)). Furthermore, smaller CRM size increases boundary layer cloud amounts independent of the assumed orientation. These configurational dependencies are important to characterize further EMAC model simulations using a similar kind of CRM setups.

The usage of superparametrised GCMs is still highly computational expensive (factor 15 to 45 increase in CPU time using 16 to 64 cells in a 2D orientation; factor 40 to 120 using the full 3D setup with 16 to 64 cells for EMAC simulations), and it is thereby desirable to use as few as possible resources without significantly modifying the model performance. All in all, it is recommended to use at least 32 cells for any setup of the superparametrisation and even proportionally more if sub-kilometre CRM grid sizes are applied. Furthermore, based on the performed analysis it is assumed that increasing the GCMs resolution to grid spacing between 50 to 100 km and successively adapting the CRM domain could lead to unexpected results because CRM statistics influence the mean climate state. Further research is required to fully answer the effect of changing the GCM grid size (i.e. modified CRM forcing) within a superparameterisation framework as proposed by Heinze et al. (2017b). In particular, it seems that cloud evolution inside of the CRM is prevented using 16 or less cells, thereby it is necessary to establish the communication across GCM cells (Arakawa et al., 2011; Jung and Arakawa, 2010).

This work has specifically been constructed to diagnose the horizontal configuration of the embedded CRM neglecting the possibility to adapt the vertical resolution. This issue has been demonstrated in Parishani et al. (2017) improving the representation of boundary layer clouds with increased vertical resolution. Most of the past research concerning superparameterisations has assumed that the vertical grids of the CRM and GCM are the same. Only recently a regional superparameterisation has been developed accounting different vertical grids but still focusing on shallow cumulus clouds (Jansson et al., 2019). Nevertheless, more research is required in this field, because most studies neglect the potential to improve mid-level cloudiness using a higher vertical resolution (Stevens et al., 2020).

In conclusion, a last point has to be taken into account that deals with the almost neglected tuning process of the superparametrised version of EMAC. The control simulation has undergone several stages of tuning activities by the model developers and specific tuning efforts (Mauritsen et al., 2012). In order to optimise a GCM, thousands of model runs are required to cover the complete parametric space of tunable variables. In addition to that, multiple process- or target-oriented constraints should be used to achieve a best model estimate for present-day climatology (Hourdin et al., 2017). Within this study the only limitation has been the energy balance at the top of the atmosphere. Future studies should, for the time being, focus on tuning this version of EMAC to multiple observational data sets, especially aiming attention at total cloud cover, which is currently underestimated (Marchand and Ackerman, 2010). Because of the high computational expense it would be advantageous to use shorter hindcast simulations with an automatic tuning in order to accelerate the progress of the superparametrised version of EMAC (Zhang et al., 2018).

The modular framework of MESSy provides an optimal model structure to easily couple the superparametrisation with other submodels. First steps have been taken to adapt cloud optical calculations and the radiative transfer scheme to be applied with subgrid-scale outputs of the CRM. Other future studies should deal with transporting chemical tracers within the superparametrisation. This would give new insights when evaluating the subgrid-scale transport of various trace gases and their diverse atmospheric lifetimes in comparison to GCM transport routines using parametrised massfluxes to describe the vertical transport.

*Code availability.* The Modular Earth Submodel System (MESSy) is continuously further developed and applied by a consortium of institutions. The usage of MESSy and access to the source code is licensed to all affiliates of institutions which are members of the MESSy Consortium. Institutions can become a member of the MESSy Consortium by signing the MESSy Memorandum of Understanding. More information can be found on the MESSy Consortium Website (www.messy-interface.org).
If eligible access can be granted to the model source code under zenodo (Rybka and Tost, 2019b). The scripts to analyse the simulations and generate the same kind of figures are archived under zenodo as well (Rybka and Tost, 2019a).
The code for using the superparametrised version of EMAC used in this manuscript will be included in the next official MESSy version (v2.55).

*Data availability.* All presented datasets are freely available. If possible DOIs or corresponding websites have been issued containing a brief summary of the corresponding dataset and a link to the data access page.

*Author contributions.* HR and HT designed the experiments. HR implemented the model code, performed the simulations and analysed the data. HR wrote the manuscript with support from HT. HT supervised the project.

*Competing interests.* The authors declare that they have no conflict of interest.

*Acknowledgements.* Parts of this research were conducted using the supercomputer Mogon and/or advisory services offered by Johannes Gutenberg University Mainz (https://hpc.uni-mainz.de), which is a member of the AHRP (Alliance for High Performance Computing in Rhineland Palatinate, www.ahrp.info) and the Gauss Alliance e.V. The authors gratefully acknowledge the computing time granted on the supercomputer Mogon at Johannes Gutenberg University Mainz.

The authors would like to thank the two anonymous reviewers for their helpful comments on the discussion paper.

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
