# Peer review of "Superparameterised cloud effects in the GCM EMAC (v2.50) - influences of model configuration"

_Geoscientific Model Development, 2019_

## Short Comment (SC1) · 4 Sep 2019

Dear authors,

Regarding your statement in the *Code availability section* "As the MESSy code is only available under license, no DOI is possible for MESSy code versions."

Please note, that DOI and license restrictions do not contradict each other. You can acquire a DOI for your code, but you do not need to make it publicly available. See for example https://doi.org/10.5194/gmd-2019-14 . This is a very similar case, where

authors are not allowed to give the code (here the COSMO model) away before certain license requirements are met.

Thus there is no reason to not provide a DOI for the code used for the results in your article.

Best regards,

Astrid Kerkweg (Executive Editor)

————————————————————

---

## Short Comment (SC2) · 4 Sep 2019

The DOI of the good example is wrong the correct one is: https://doi.org/10.5194/gmd-2019-141 In the copy process the last 1 got missing.

Sorry for the confusion!

Best regards, Astrid Kerkweg
* * *
[Figure]

2019.

---

## Author Comment (AC1) · 5 Sep 2019

Dear Executive Editor,

thank your for mentioning the possibility to provide a source code DOI with a restricted access. The source code and necessary input files have been uploaded to zenodo archive (https://doi.org/10.5281/zenodo.3386969). Access will be granted if license requirements are met.

In addition to that scripts for analysis and plotting are provided as an open source archive as well (https://doi.org/10.5281/zenodo.3387004).

The section 'Code availability' will be updated with these references after the discussion with the reviewers has started.

Best regards,

Harald Rybka

---

## Referee Comment (RC1) · Anonymous Referee #1 · 24 Sep 2019

This manuscript evaluates the impact of various configurations of the embedded cloud-resolving model in the ECHAM climate model on a relatively short climate simulation when compared to various climate-relevant observations and reanalysis. Overall, this is a worthy effort and it should be published. However, I feel there are many issues (both minor technical and more significant general) that need to be addressed before the paper is accepted.

General comment:

[Figure]

U.S. National Science Foundation supported the Science and Technology Center called Center for Modeling of Atmospheric Processes (CMMAP) between 2006 and 2016, see http://saddleback.atmos.colostate.edu/cmmap/. There is an extensive list of publications produced by CMMAP at http://saddleback.atmos.colostate.edu/cmmap/research/pubs-ref.html that the authors of the paper under review may find useful for the motivation of their investigation. I vaguely remember that some of the superparameterization (SP) tests reported in the current paper were also tried by the people involved in CMMAP (e.g., M. Khairoutdinov, M. Pritchard). Perhaps such efforts should be mentioned in the current manuscript and some of the outcomes can be compared.

Specific comments.

1. I found the title of the paper awkward. First, 15-month simulations cannot be considered long from the climate perspective. Second, the two parts of the title are poorly linked. Please revise.

2. P2L23 (page 2, line 23) and in couple other places in the manuscript: it is not clear to me what is meant by "embedding an ensemble of interacting CRMs". Only a single CRM is embedded in each climate simulation, correct? And the configuration is changed in different simulations, correct? If so, referring to an ensemble of simulations is confusing. Please revise.

3. P2L30: "drastically reduced". First, the cost depends on the configuration. According to M. Khaiouritdinov, the initial implementation of SAM in CAM as reported in 2001 GRL paper slowed down CAM about 200 times. For "larger" CRM (i.e., more columns or higher resolution that increase CRM effort) this number should increase. The dependence on the number of CRM columns should also be valid for the 3D CRM. That said, there are also obvious benefits of separating small-scale and large-scale dynamics, such as parallelization, what model equations to use, etc. Grabowski (JMSJ 2016, p. 327, "Towards global large eddy simulation: super-parameterization revisited") dis-

cusses some of these issues.

4. P3L1: A reference to CMMAP would be appropriate here. A selection of papers from the CMMAP website can be used in this paragraph.

5. Fig. 1. First, it shows two timesteps, not three as stated in P5L7, correct? Second, the coupling terms shown in the figure should be shown in text with appropriate reference (e.g., Grabowski JAS 2006). Is momentum coupled as well as thermodynamic fields?

6. P5L17: Why is radiation singled out here? What about surface fluxes and boundary layer transports? What about the land-surface model? Ocean SST? Please explain clearly which processes are treated by the GCM and which by CRMs.

7. Table 1. I vaguely remember that Dr. Khairoutdinov conducted similar tests to some of those included in the table. Perhaps he can point the authors to results of some of those tests.

8. P6L5: "ensemble: - see 2 above.

9. P6L9. Sending reader to the supplement is not appropriate. At least some basic features of the simulation setup should be mentioned here.

10. P7L2. 15 months is pretty short for climate simulations. How robust are results reported in the paper?

11. P7L26. I am sure Khairoutdinov ran and reported results from small 3D CRM setup in SP CAM. Again, referring to his experience with this extremely small domain would be needed here.

12. P7L30. Please explain how momenta are coupled. See 5 above.

13. Table 2. It would be great to have some error bars for all entries in the table. For the observations, annual variability can provide that, correct? The same could be done for multiyear simulations, except that the simulations are short. This is an important

aspect and it requires some comment and maybe additional simulations.

14. P10L8. For a fair comparison between EMAC and SP-EMAC, one needs to ensure that cloud radiative properties are prescribed as closely as possible. Please explain this element of the model setup. Is this included in the supplement? See 9 above.

15. P10L29. It si not clear to me how a 15-month simulation can be compared to the reanalysis. Perhaps it can if the setup is designed appropriately. Please explain.

16. Figure 2 (and maybe other figures). I suggest not to use a color for CTRL, but a symbol (e.g., a star). This would allow CTRL to better stand out.

17. P12L4. "Thereby almost no water vapor...". I do not see the link between this sentence and the previous one. Either way, is this really correct?

18. Figure 5. Are the differences statistically significant? See 13 above. Also, maps in the right panel show very little variability. Are they needed?

19. P14L15: "The most distinct...". Looking at the figure, I am not sure what the authors have in mind here.

20. P15L23: A reference to Guichard et al. (QJ 2004) would be also appropriate here.

21. I feel one should also mention vertical resolution (both in a GCM and in CRM) as a potential factor affecting model results. This should be brought somewhere in the paper.

22. P17 and Fig. 7. Problems with radiative fluxes over the Southern Ocean are well appreciated by the climate community. This region was targeted in recent field campaigns (e.g., SOCRATES, see https://www.eol.ucar.edu/field_projects/socrates). I think the scientific consensus is that the representation of cloud microphysical processes such as partitioning between water and ice is an important factor. Can SAM's rather poor microphysics cope with this issue? Should this aspect be mentioned in the discussion?

23. P18L12: The sentence "Therefore it not appropriate..." comes out of nowhere! Is part of this sentence missing? Please revise or explain what specifically is meant here.

24. P19L14. Allowing CRMs to rotate was first applied in Grabowski (JAS 2004, p.1940). This reference should be added here. Is the model vertical resolution relevant to the problem discussed in this section? I would think so.

24. Figure 8. What is the reason for the noise evident in CRE_SW SP-EMAC (and leading to noise for NetCRE)? This noise is also noticeable in CTRL simulations.

---

## Referee Comment (RC2) · Anonymous Referee #2 · 22 Oct 2019

General comments:

This paper systematically presents climatological simulation results of EMAC with super-parameterization with 20 various configurations, including CRMs orientation, cell size and number of cells. It is useful and impressive to see these results as in Table 2. This paper is publishable if clearer messages are provided.

For the current version, the readers only know from the abstract that only some aspects of tropical precipitation are better represented with the super-parameterized EMAC

compared with the CTRL with a convectional convection parameterization. The other aspects depend on the choice of the CRM setup, and the super-parameterized simulations deteriorate in some cases. What the readers want to know are whether and when the super-parameterized EMAC becomes better than CTRL, and what kind of suitable setup should be chosen for CRMs.

One interesting result is that results of the super-parameterized simulations are divided into two groups (sub-ensemble A/B). More analysis is suggested to explain why this separation exists. It seems that the results do not clearly depend on the CRM setup.

It is suggested that effects of momentum transport should be summarized in the abstract.

The authors also should argue about the similarities and the differences of the general behaviors of the effect of the super-parameterization with previous studies of the other groups (Khairoutdinov, W-K Tao, etc.).

Specific comments:

p. 1, L11, "cloud cover": This is ambiguous. Is this high cloud fraction or total cloud fraction? Needs a clear definition.

p. 1, L12, "hydrological overturning is too efficient": This is not clear and may cause confusion. What is the meaning of "efficient circulation"? The authors might want to mention precipitation efficiency. However, in this case, it is not straightforward to related cloud fraction and precipitation efficiency. Even precipitation efficiency is unchanged, cloud cover may decrease if cloud thickness decreases.

p. 1, L15, "diurnal cycle of precipitation": In general, diurnal cycle of precipitation is reasonably captured by CRMs. One may think that if "diurnal cycle of precipitation" is not properly simulated, something wrong in parameter settings of CRMs.

p. 3, L8-9, "To our knowledge this is the first attempt summarizing the effects of different configurations of the super-parameterization onto the model mean climate state."

I think that there exist similar studies on the effects of different cloud microphysics schemes on the model climate.

p. 9, Table 2: precipitable water should be added and discussed.

p. 10, L6, "observed value": Please add a reference to this value.

p. 10, L19, "All simulations show shortwave and net radiative fluxes at TOA that are in close agreement to observed fields": This is not clear whether the SP-EMAC runs are better than CTRL.

p. 10, L22-23: Any difference between "cloud cover" and "cloud amount"? Is "cloud cover" of SP-EMAC generally better than CTRL? Why?

p. 10, L35, "The overestimated variability of specific humidity is mainly a cause of too much water vapor transport over tropical continents and too less over tropical oceans": Is this general behavior of SP-EMAC? Why?

p. 12, L11-15: From this summary paragraph, it is not clear the real advantage of the super-parameterizaion. Is this the correct message of this paper?

p. 13, L5-11: Related to the discussion around this paragraph, the authors should refer to Luo and Stephens (2006), "An enhanced convection-wind-evaporation feedback in a superparameterization GCM (SP-GCM) depiction of the Asian summer monsoon" (Geophys. Res. Lett., 33, L06707, doi:10.1029/2005GL025060).

p. 14 L3-6: The logic of this paragraph is not clear. Why the super-parameterization affects land-ocean contrast?

p. 15, L15-17: It is suggested that Figure 3 should be compared with the zonal mean precipitation of global cloud-resolving models by Stevens et al. (2019, Fig. 5, https://doi.org/10.1186/s40645-019-0304-z).

p. 16, L4: Which product of TRMM is used. Refer to Sato et al. (2009), "Diurnal cycle of precipitation over the tropics simulated by a global cloud resolving model." (J. Clim.,

22, 4809-4826, doi:10.1175/2009JCLI2890.1).

p. 18, Section 3.3: Please discuss robustness of the difference between the sub-ensemble A and B. It seems that the difference is not systematically depend on the CRM configuration. Can the authors say in which cases the category of the sub-ensemble is determined.

p. 19, L28-29: Why the results are very different from MODIS?

———————————————

---

## Author Comment (AC2) · 27 Jan 2020

**Authors response to review number 1:**

Review comments pasted in black.
Author response in blue.
IMPORTANT NOTIFICATION (in red)!

[Figure]

This manuscript evaluates the impact of various configurations of the embedded cloud-resolving model in the ECHAM climate model on a relatively short climate simulation when compared to various climate-relevant observations and reanalysis. Overall, this is a worthy effort and it should be published. However, I feel there are many issues (both minor technical and more significant general) that need to be addressed before the paper is accepted.

We would like to thank the reviewer for their helpful report. The manuscript has been revised according to the referees comments.

One important thing to mention is that we have found a bug (precipitation fluxes have been set to zero after CRM has been called) in almost two third of the SP-EMAC simulations and re-simulated these. Almost all results are affected, except the precipitation PDFs and diurnal cycles of rainfall in chapter 3.2 (Fig. 5 and 6). Nevertheless, most of the important features did not change. Taylor diagrams (Fig. 2) changed slightly.

Precipitation biases (Fig. 3 and 4) did not change significantly as well as cloud radiative effects (Fig. 7 and 8).

Chapter 3.3 has been rewritten, because the separation in a Sub-Ensemble A and B was caused by this error in the code. Straightaway, all (latent and sensible) heatfluxes are in agreement with NCEP reanalysis. An additional analysis has been performed in section 3.3 to evaluate the issue of CRM configuration onto the simulated cloud amount in SP-EMA.

Most of the reviewers comments concerning chapter 3.3 are referred to this bug-fix.

General comment: U.S. National Science Foundation supported the Science and Technology Center called Center for Modeling of Atmospheric Processes (CMMAP) between 2006 and 2016, see http://saddleback.atmos.colostate.edu/cmmap/. There is an extensive list of publications produced by CMMAP at http://saddleback.atmos.colostate.edu/cmmap/research/pubs-ref.html that the authors

of the paper under review may find useful for the motivation of their investigation. I vaguely remember that some of the superparameterization (SP) tests reported in the current paper were also tried by the people involved in CMMAP (e.g., M. Khairoutdinov, M. Pritchard). Perhaps such efforts should be mentioned in the current manuscript and some of the outcomes can be compared.

Thank you for mentioning CMMAP which we have been aware of. We included some references of this project within the introduction and mentioned findings of particular publications where comparing their outcomes with ours.

Specific comments.

1. I found the title of the paper awkward. First, 15-month simulations cannot be considered long from the climate perspective. Second, the two parts of the title are poorly linked. Please revise.

We have revised the title to avoid confusion and thereby specified the main focus of this paper. The new title is: 'Superparameterised cloud effects in the GCM EMAC (v2.50) - influences of model configuration'

2. P2L23 (page 2, line 23) and in couple other places in the manuscript: it is not clear to me what is meant by "embedding an ensemble of interacting CRMs". Only a single CRM is embedded in each climate simulation, correct? And the configuration is changed in different simulations, correct? If so, referring to an ensemble of simulations is confusing. Please revise.

The term "ensemble" is misleading because it is usually applied to combine different models groups or model setups. In this instance ensemble is referring to the individual grid boxes of a single CRM which are interacting within a GCM column. To avoid any possibility of confusion the term "ensemble" in connection with the CRM has been avoided and revised throughout the manuscript.

3. P2L30: "drastically reduced". First, the cost depends on the configuration. According to M. Khaiouritdinov, the initial implementation of SAM in CAM as reported in 2001 GRL paper slowed down CAM about 200 times. For "larger" CRM (i.e., more columns or higher resolution that increase CRM effort) this number should increase. The dependence on the number of CRM columns should also be valid for the 3D CRM. That said, there are also obvious benefits of separating small-scale and large-scale dynamics, such as parallelization, what model equations to use, etc. Grabowski (JMSJ 2016,p. 327, "Towards global large eddy simulation: super-parameterization revisited") discusses some of these issues.

The cost to run a model including a superparameterisation is strongly dependent on the number of columns of the embedded CRM. Even the grid size of a single CRM grid box has an effect on the CPU time, i.e. keeping the number of CRM cells within each GCM grid box constant and lowering the resolution increase the CPU time. This is due to small instabilities, which occur more often for smaller CRM resolutions for a fixed CRM time step, within SAM which (to some degree) are automatically avoided by subsequently decreasing the CRM time step. In comparison to CAM (Khairoutdinov, 2001) EMAC has slowed down by a factor between 40 (2D orientation, 16 CRM cells, 4 km) to 120 (3D CRM, 64 cells, 1 km; depending on the CRM configuration). All in all, as stated in Grabowski (JMSJ, 2016), the conventional usage of a superparameterisation reduces the amount of computational time by approximately three orders of magnitude in comparison to global CRMs like NICAM.

4. P3L1: A reference to CMMAP would be appropriate here. A selection of papers from the CMMAP website can be used in this paragraph.

References of CMMAP have been added.

5. Fig. 1. First, it shows two timesteps, not three as stated in P5L7, correct? Second,

the coupling terms shown in the figure should be shown in text with appropriate reference (e.g., Grabowski JAS 2006). Is momentum coupled as well as thermodynamic fields?

Figure 1 shows two timesteps (three points in time - corrected in the manuscript) and the coupling terms have been referenced to Grabowski, 2006 JAS. As mentioned briefly in the text (P5L15) momentum (CRM forcing for $u$ and $v$) is coupled as well but CRM feedback (convective momentum transport) is only applied for 3D CRM cases (simulation number 15 to 20 in table 1). 2D CRM configurations neglect zonal/meridional convective momentum on the large-scale flow. This has been rephrased to avoid ambiguities.

6. P5L17: Why is radiation singled out here? What about surface fluxes and boundary layer transports? What about the land-surface model? Ocean SST? Please explain clearly which processes are treated by the GCM and which by CRMs.

In order to consider subgrid-scale clouds and their radiative feedback on the subgrid-scale the radiation code as well as the cloud optical properties have to be modified to run on the CRM grid. This was done after the implementation of the superparameterisation was succesful. In order to account for model differences due to the usage of a superparameterisation instead of using cloud and convection parameterisations we chose to only switch these parts of the model. A further modification concerning cloud optical properties and radiative transfer would complicate the analysis to differentiate model discrepancies between SP-EMAC and CTRL. Differences could be either due to a different cloud development within the superparameterisation or cloud radiative effects considering subgrid-scale cloud fractions.

SST is prescribed and is not changed by the superparameterisaion. Surface fluxes as well as boundary layer transport is done on the GCM grid based on formulations of Roeckner (2003; https://www.mpimet.mpg.de/fileadmin/publikationen/Reports/max_scirep_349.pd).

The section of the model description has been expanded to account for these processes within the description.

7. Table 1. I vaguely remember that Dr. Khairoutdinov conducted similar tests to some of those included in the table. Perhaps he can point the authors to results of some of those tests.
We have modified table 1 marking all SP-EMAC configurations which have been used in previous literature. Most of these simulations are difficult to compare with ours because different vertical grid spacings or further modifications of microphysics, turbulence or radiation have been added. In addition to that, many simulations cover even shorter time periods (couple of months). Nevertheless, results are compared in the context of these simulations.

8. P6L5: "ensemble: - see 2 above.
See answer 2 above.

9. P6L9. Sending reader to the supplement is not appropriate. At least some basic features of the simulation setup should be mentioned here.
The most important characteristics for all simulations are listed within this paragraph: resolution, time period, initial conditions and parametrisations used. The supplement provides only additional information to accurately repeat the simulations. Some extra information has been added in terms of: surface fluxes, boundary layer transport, surface scheme, cloud optical properties and radiation. See answer 6 and 14.

10. P7L2. 15 months is pretty short for climate simulations. How robust are results reported in the paper?
In order to evaluate robustness of our results two SP-EMAC simulations have been elongated to provide interannual variability which is be presented in table 2 in the new

manuscript. All in all, the results for the 15 month period are robust with respect to the two multi-year long simulations.

11. P7L26. I am sure Khairoutdinov ran and reported results from small 3D CRM setup in SP CAM. Again, referring to his experience with this extremely small domain would be needed here.

These edge case scenarios covering very small CRM domain sizes within a GCM grid cell have not been reported by Dr. Khairoutdinov. An exception is the work of Parishani (JAMES, 2017), who embedded a 3D CRM but with a much higher horizontal (250m) and vertical (down to 20m) resolution. This research focused only on shallow cumulus boundary layer clouds where small domain sizes have successfully been employed (Ackermann, GRL, 2003). Domain sizes are far more restrictive for deep convection and associated high clouds, where mesoscale organization and cold pools play a crucial role for its development.

The paragraph concerning very small domain sizes has been revised.

12. P7L30. Please explain how momenta are coupled. See 5 above.

See answer 5 above.

13. Table 2. It would be great to have some error bars for all entries in the table. For the observations, annual variability can provide that, correct? The same could be done for multiyear simulations, except that the simulations are short. This is an important aspect and it requires some comment and maybe additional simulations.

Error bars for observations have been included as uncertainty in table 2. This uncertainty includes interannual variability over the observational time period as well as measurement uncertainty given within the observational descriptions. Providing interannual variability for all simulations is not possible because of too much computational time. To provide an estimate of interannual variability, two simulations are further

integrated to achieve a multiyear simulation (10 years). The annual variability will be given in table 2 as well providing exact model configuration which has been chosen for the multiyear simulation.

14. P10L8. For a fair comparison between EMAC and SP-EMAC, one needs to ensure that cloud radiative properties are prescribed as closely as possible. Please explain this element of the model setup. Is this included in the supplement? See 9 above.

Cloud radiative properties are treated in a similar manner for all simulations using the submodel CLOUDOPT (Dietmueller, GMD, 2016, section 2.4; doi:10.5194/gmd-9-2209-2016). In order to calculate cloud optical properties the following input variables are necessary: cloud cover, cloud water content, ice water content, cloud nuclei concentration whereas CNC has a fixed exponentially decreasing profile for land and ocean separately. Based upon LWC and IWC effective radii are calculated (Johnson, 1993; Moss, 1996). In order to account for cloud overlap the assumption of maximum-random overlap has been chosen.

The technical aspects and theory is described in Dietmueller, 2016. For our purpose the important aspects are:

- model resolution-dependent parameters (asymmetry factor and cloud inhomo-geneity) kept fixed for all simulations based on the T42 resolution of the host GCM

- channel objects are different for CTRL and SP-EMAC:

  – for CTRL cloud water content, cloud ice content, cloud cover are large-scale variables calculated within the CLOUD submodule (Roeckner, 2006)

  – for SP-EMAC cloud cover, cloud water and ice are calculated as horizontal means over all CRM grid boxes within a GCM column. Thereby no subgrid-scale calculation of cloud optical properties (as well as radiative tendencies)

is done. This method has been applied intentionally to use subgrid-scale information of CRM but condense it onto the coarse GCM grid and use the same submodels.

Our SP-EMAC simulations are equivalent to Cole, 2005 (doi: 10.1029/2004GL020945) experiment number 3. This type of experiment ensures that the only difference between CTRL and SP-EMAC is the substition of the Tiedtke convection and large-scale cloud parameterisation with the embedded CRM as a superparameterisaion.
An additional paragraph has been added to section 2.3 explaining the coupling. See answer to question 6.

15. P10L29. It si not clear to me how a 15-month simulation can be compared to the reanalysis. Perhaps it can if the setup is designed appropriately. Please explain.
The simulations are designed to represent a simulated climatological reference to prescribed SST/SIC and greenhouse gas distributions. NCEP Reanalysis is used as quasi-observations to compare atmospheric quantities at different heights (here: specific humidity) to evaluate the new model ability to capture the distribution and variability. A short sentence has been added to clarify this.

16. Figure 2 (and maybe other figures). I suggest not to use a color for CTRL, but a symbol (e.g., a star). This would allow CTRL to better stand out.
Using a different symbol for CTRL in all Taylor diagrams would be preferable. The problem is that for some Taylor diagrams multiple quantities are shown separated via different symbols (e.g. Figure 2 - radiation, specific humidity and Figure 9) therefore it would confuse the reader to see the same symbol for CTRL indicating different quantities. Using a dark purple color is the best choice in our opinion.

17. P12L4. "Thereby almost no water vapor. . .". I do not see the link between this

sentence and the previous one. Either way, is this really correct?

The link between the unresolved stratospheric circulation and the transport of water vapour is the Brewer-Dobson circulation. Because the vertical resolution within the lower stratosphere and upper troposphere (as well as the uppermost model layer which is located at 10 hPa) is too coarse to explicitly resolve the large-scale Brewer-Dobson circulation no water vapor is transported from tropics to the poles. The origin of higher water vapour concentrations at the tropical UTLS is mostly convectively produced via vertical transport (updrafts) of moist air from the lower troposphere.

The sentence has been rephrased.

18. Figure 5. Are the differences statistically significant? See 13 above. Also, maps in the right panel show very little variability. Are they needed?

The differences in figure 5 are statistically significant. In conjunction with figure 4, which shows only regions which are different on a significance level of 90 %, the Warm Pool region and southern ocean mid-latitudes have been chosen because they reveal the most distinct differences in precipitation rates. Even if multiyear averages (for EMAC CTRL) are taken into account these regions show significant differences in comparison with GPCP (Tost et al, 2006, ACP).

Concerning the maps in the right panel showing very little variability: This is primarily due to the range of the color bar used to produce these maps. This will be updated to a smaller range in order to better visualize the variability.

19. P14L15: "The most distinct. . .". Looking at the figure, I am not sure what the authors have in mind here.

Looking at the PDF of monthly precipitation for the maritime continent in figure 5 it is evident that some individual SP-EMAC simulations (not shown) are very close to the GPCP data. This can be stated as a most distinct feature because the variability (grey shaded area displaying the range of all superparameterised simulations) of all

SP-EMAC simulations covers almost the entire range of observed precipitation rates (purple line).

20. P15L23: A reference to Guichard et al. (QJ 2004) would be also appropriate here. Added this reference.

21. I feel one should also mention vertical resolution (both in a GCM and in CRM) as a potential factor affecting model results. This should be brought somewhere in the paper.
A paragraph concerning vertical resolution has been added within chapter 3.3 as well as in the discussion. See question 24.

22. P17 and Fig. 7. Problems with radiative fluxes over the Southern Ocean are well appreciated by the climate community. This region was targeted in recent field campaigns (e.g., SOCRATES, see https://www.eol.ucar.edu/field_projects/socrates). I think the scientific consensus is that the representation of cloud microphysical processes such as partitioning between water and ice is an important factor. Can SAM's rather poor microphysics cope with this issue? Should this aspect be mentioned in the discussion?
This issue should be mentioned in the discussion. The simple microphysics within SAM in conjunction with the relatively low vertical resolution is an important factor controlling the CRE in these regions. Reducing this problem can be achieved by using a higher vertical resolution of the embedded CRM (Parishani, 2017, JAMES) as well as using a two-moment microphysical scheme and a higher-order turbulence scheme (Wang, 2017; https://doi.org/10.1002/2014MS000375).
A short paragraph has been added in the discussion.

23. P18L12: The sentence "Therefore it not appropriate. . ." comes out of nowhere! Is part of this sentence missing? Please revise or explain what specifically is meant here.

The sentence has been rephrased. —"Future studies with SP-EMAC should always look at the different cloud radiative effects to avoid misinterpretations of model results. This is necessary because not all SP-EMAC configurations are appropriate to use and addresses the need for a tuning activity for SP-EMAC in the near future."

24. P19L14. Allowing CRMs to rotate was first applied in Grabowski (JAS 2004, p.1940). This reference should be added here. Is the model vertical resolution relevant to the problem discussed in this section? I would think so.

Reference has been added. The vertical resolution plays an important role considering boundary layer clouds especially for the stratocumulus region (Parishani, JAMES, 2017). Another point is that model layer thickness at the tropopause and within the stratosphere is very coarse to reproduce the transport of water vapor through penetrating deep convective clouds realistically. Related to that is the explicit representation of cirrus clouds which is hardly possible using a vertical resolution on the order of $\mathcal{O}(3)$ metres.

25. Figure 8. What is the reason for the noise evident in CRE_SW SP-EMAC (and leading to noise for NetCRE)? This noise is also noticeable in CTRL simulations.

The reason for the wave-like structure ("noise") most prominent in the Pacific is due to the spectral core of the host model EMAC.

---

## Author Comment (AC3) · 27 Jan 2020

**Authors response to review number 2:**

Review comments pasted in black.
Author response in blue.
IMPORTANT NOTIFICATION (in red)!

General comments:
This paper systematically presents climatological simulation results of EMAC with super-parameterization with 20 various configurations, including CRMs orientation, cell size and number of cells. It is useful and impressive to see these results as in Table 2. This paper is publishable if clearer messages are provided.

We would like to thank the reviewer for their helpful report. The manuscript has been revised according to the referees comments.

One important thing to mention is that we have found a bug (precipitation fluxes have been set to zero after CRM has been called) in almost two third of the SP-EMAC simulations and re-simulated these. Almost all results are affected, except the precipitation PDFs and diurnal cycles of rainfall in chapter 3.2 (Fig. 5 and 6). Nevertheless, most of the important features did not change. Taylor diagrams (Fig. 2) changed slightly.

Precipitation biases (Fig. 3 and 4) did not change significantly as well as cloud radiative effects (Fig. 7 and 8).

Chapter 3.3 has been rewritten, because the separation in a Sub-Ensemble A and B was caused by this error in the code. Straightaway, all (latent and sensible) heatfluxes are in agreement with NCEP reanalysis. An additional analysis has been performed in section 3.3 to evaluate the issue of CRM configuration onto the simulated cloud amount in SP-EMA.

Most of the reviewers comments concerning chapter 3.3 are referred to this bug-fix.

For the current version, the readers only know from the abstract that only some aspects of tropical precipitation are better represented with the super-parameterized EMAC compared with the CTRL with a convectional convection parameterization. The other aspects depend on the choice of the CRM setup, and the super-parameterized simulations deteriorate in some cases. What the readers want to know are whether and when the super-parameterized EMAC becomes better than CTRL, and what kind of suitable

setup should be chosen for CRMs.

The abstract has been modified to include the best suited setup for SP-EMAC as well as more detailed description of advantages and disadvantages using the superparameterisation.

One interesting result is that results of the super-parameterized simulations are divided into two groups (sub-ensemble A/B). More analysis is suggested to explain why this separation exists. It seems that the results do not clearly depend on the CRM setup.

Due to a bug this division in two sub-ensembles had happened (see important notification at the beginning of our response). Section 3.3. has been completely modified to emphasize other difference of the superparameterisation due to specific configurations.

It is suggested that effects of momentum transport should be summarized in the abstract. In order to analyse the disitinct effect of momentum transport further simulations are needed to allowing/preventing the momentum transport for each configuration. This issue is out of our scope but could be evaluated in a future study.

The authors also should argue about the similarities and the differences of the general behaviors of the effect of the super-parameterization with previous studies of the other groups (Khairoutdinov, W-K Tao, etc.).

The manuscript has been revised to include more comparisons with previous studies (this point has been mentioned by reviewer number 1 as well).

Specific comments:
p. 1, L11, "cloud cover": This is ambiguous. Is this high cloud fraction or total cloud fraction? Needs a clear definition.

"cloud cover" changed to "total cloud cover".

p. 1, L12, "hydrological overturning is too efficient": This is not clear and may cause confusion. What is the meaning of "efficient circulation"? The authors might want to mention precipitation efficiency. However, in this case, it is not straightforward to related cloud fraction and precipitation efficiency. Even precipitation efficiency is unchanged, cloud cover may decrease if cloud thickness decreases.
We have modified this terminology to avoid confusion. The systematic underestimation is related to the "untuned" version of SP-EMAC. Instead of linking total cloud cover to precipitation efficiency, section 3.3 reveals in-atmosphere cloud amount to certain CRM configurations. Thereby the sensitivity of SP-EMAC configuration on total cloud cover is estimated and recommendations are provided for future studies.

p. 1, L15, "diurnal cycle of precipitation": In general, diurnal cycle of precipitation is reasonably captured by CRMs. One may think that if "diurnal cycle of precipitation" is not properly simulated, something wrong in parameter settings of CRMs.
This is in general correct but for all superparameterised simulations the diurnal cycle (over land) shows a similar onset in precipitation during the morning hours. This feature is totally misrepresented by using a convection parameterisation. The amplitude of diurnal precipitation exposes a dependency on the number of CRM cells. This is not related to wrong parameter settings but to a prohibited cloud development because the CRM domain becomes too small when decreasing the number of CRM cells.

p. 3, L8-9, "To our knowledge this is the first attempt summarizing the effects of different configurations of the super-parameterization onto the model mean climate state." I think that there exist similar studies on the effects of different cloud microphysics schemes on the model climate.
Of course there exists multiple publications concerning new setups of superparameterisation especially including different microphysical scheme. A few studies show different setups for superparameterisations (Khairoutdinov, 2005, Marchand and Ackermann, 2010, Pritchard and Sommerville, 2009). We have included within table 1 which configuration has been used in previous literature. Nevertheless no study has evaluated the total impact of CRM configuration in such a manner for a simulated climatological year.

p. 9, Table 2: precipitable water should be added and discussed.
A few sentences have been added within the paragraph to integrate total precipitable water in the context of global mean values.

p. 10, L6, "observed value": Please add a reference to this value.
The reference to the CERES data has been added as well as the reference to table 1 including the exact observational data set used for comparison.

p. 10, L19, "All simulations show shortwave and net radiative fluxes at TOA that are in close agreement to observed fields": This is not clear whether the SP-EMAC runs are better than CTRL.
That is correct. When comparing SP-EMAC with CTRL no significant improvement of the shortwave or net radiative flux at TOA can be deduced. But globally speaking a very similar skill is achieved compared to CERES data. This paragraph has sligthly been modified to specify this.

p. 10, L22-23: Any difference between "cloud cover" and "cloud amount"? Is "cloud cover" of SP-EMAC generally better than CTRL? Why?
When speaking of "cloud cover" the total cloud cover within a GCM grid column is meant, i.e. it is the two-dimensional representation of cloud coverage including cloud

overlap assumptions. In-atmosphere "cloud amount" refers to the cloud cover within a GCM grid box, i.e. three-dimensional cloud cover variable. This has been changed in the manuscript.

As mentioned within the paper the total cloud cover in comparison with CTRL is in improved in terms of spatial and temporal variability. In particular the northern hemisphere as well as tropical land shows an improvement for total cloud cover. Oceanic regions reflect an underestimation of total cloud cover. This is predominantly related to the too early onset on precipitation for maritime (almost pure liquid) clouds thereby removing most of the condensate from the atmosphere. In order to improve shallow boundary layer clouds a higher order microphysical scheme is needed and ultimately a finer (vertical) resolution. The improvement for continental clouds is mostly due to a better partitioning of liquid and ice clouds. This is related to the dynamical aspect of subgrid-scale cloud development within the CRM representing a better cloud evolution in terms of updrafts and downdrafts. This allows water vapour to be transported into the correct heights for condensation.

These aspects have been included within the specific sections in a new version of the paper.

p. 10, L35, "The overestimated variability of specific humidity is mainly a cause of too much water vapor transport over tropical continents and too less over tropical oceans": Is this general behavior of SP-EMAC? Why?

This effect is not only visible in SP-EMAC. The control simulation shows a similar behavior with parametrised convection. Especially one configuration (OR2 4km 64) shows a almost perfect representation of water vapour distribution at 250 and 500 hPa (see Fig. 2, Taylor diagram for specific humidity green triangle and square). Therefore not all SP-EMAC simulations show this behaviour. In order to evaluate this aspect in more detail further sensitivity simulations are needed which is out of scope for our research.

p. 12, L11-15: From this summary paragraph, it is not clear the real advantage of the super-parameterizaion. Is this the correct message of this paper?
This paragraph should not emphasize the advantages of using SP-EMAC. Regarding global statistics of the evaluated variables it performs as good as CTRL with some improvements as well as a slight deterioration for precipitation. This efforts are achieved without the focus on tuning this model setup. Therefore it can be assumed that with additional tuning efforts SP-EMAC would outperform CTRL.

p. 13, L5-11: Related to the discussion around this paragraph, the authors should refer to Luo and Stephens (2006), "An enhanced convection-wind-evaporation feedback in a superparameterization GCM (SP-GCM) depiction of the Asian summer monsoon" (Geophys. Res. Lett., 33, L06707, doi:10.1029/2005GL025060).
The reference to Luo and Stephens 2006 has been added mentioning the importance of feedback mechanisms.

p. 14 L3-6: The logic of this paragraph is not clear. Why the super-parameterization affects land-ocean contrast?
It is the other way around. Land-ocean contrasts affect precipitation rates of the superparameterisation depending on the chosen CRM setup. Changing the orientation, size or number (all parameters for the CRM) affects precipitation rates especially above oceans and coastal regions within the ITCZ and northern and southern mid-latitudes inducing a high sensitivity for these regions. We suppose that regarding oceanic rainfall the setup should be carefully chosen because this could degrade the simulation results. For coastal regions we speculate that the orientation of the CRM has an effect because summarized effects of changes in roughness length, higher wind speeds (sea breeze) and evaporation is influencing the precipitation within the superparameterization depending on the chosen orientation.

p. 15, L15-17: It is suggested that Figure 3 should be compared with the zonal mean precipitation of global cloud-resolving models by Stevens et al. (2019, Fig. 5, https://doi.org/10.1186/s40645-019-0304-z).

Comparing figure 3 and GCRM zonal mean precipitation of Stevens et al. 2019 can not be easily done because the simulation period is quite different. This paper illustrate zonal means of annual averaged precipitation whereas Stevens et al. uses 30-days (of August 2016). A possibility could be to compare only August data but this would raise the question to what extent the August of 2016 represents a more or less climatological averaged month of August. We refrain comparing our results with Stevens but include the new possibilities to compare state-of-the-art GCRMs with superparameterisations within the discussion!

p. 16, L4: Which product of TRMM is used. Refer to Sato et al. (2009), "Diurnal cycle of precipitation over the tropics simulated by a global cloud resolving model." (J. Clim., 22, 4809-4826, doi:10.1175/2009JCLI2890.1).

The reference has been added. The TRMM data reflects a 12-year long period for the month of July (1998-2010). The reference for the data is available at figure 6 (see caption). This has been specified in the text.

p. 18, Section 3.3: Please discuss robustness of the difference between the sub-ensemble A and B. It seems that the difference is not systematically depend on the CRM configuration. Can the authors say in which cases the category of the sub-ensemble is determined.

This results is obsolete due to the bug-fix mentioned in the beginning. Section 3.3 has been re-written.

p. 19, L28-29: Why the results are very different from MODIS?

The results are different from MODIS observations because of neglecting sub-grid

cloud variability and the missing of COSP as a satellite simulator. As emphasized within the description of the superparameterisation setup we have adopted the modification to account for subgrid-scale cloud variability within calculation of cloud optical properties and radiation after performing all simulations. A totally fair comparison with MODIS would additionally imply the implementation of the COSP package to adapt the satellite simulator which was out of our scope. A few references and sentences to COSP and the importance of subgrid-scale cloud variability has been added to the paper (Bodas-Salcedo, 2011; Song, 2018, Swales, 2018).

---

## Referee Report (RR1)

**Review of " Superparameterised cloud effects in the GCM EMAC (v2.50) -influences of model configuration"**
**By Harald Rybka and Holger Tost**
**gmd-2019-193**

**Recommend: Minor revisions**

**General comments:**

This revised manuscript focuses the diversity of precipitation and cloud optical thickness of the superparameterized model, by fixing the bug for the simulations in the first submission. The results became more reasonable and the message of the paper became clearer. I suggest publication of this paper after fixing the points indicated below.

**Specific comments:**

p. 9, L28, "partition": It is unclear what kind of partition is stated here.

p. 11, L41, "TRMM data": More explicitly refer to the TRMM product name as "TRMM_3B42 v7". In Sato et al. (2009,JCLI), the distinction between 3B42 and 3G68 is argued, and it is stated that 3G68 is more reliable for the diurnal cycle.

p. 15, Figure 9: In this figure, the Taylor diagram of two quantities are shown: sensible and latent heat fluxes and cloud optical thickness (COT). These two quantities are not related. The left panel (sensible and latent heat fluxes) can be removed. Instead, it would be more informative if some relation exists between precipitation and COT.

p. 18, L5-9, "Nowadays state-of-the-art global cloud resolving models provide new possibilities comparing superparameterised simulations with monthly-long high resolution models (Stevens et al., 2019).": This sentence is not suitable for this paragraph. This should be moved to the introduction.

---

## Referee Report (RR2)

Review of revised manuscript "Superparameterized cloud effects in the GCM EMAC (v2.50) – influences of model configuration" by Rybka and Tost (with a modified title, previously "Effects of model configuration for superparameterized long-term simulations – Implementation of a cloud resolving model in EMAC (v2.50)".

Recommendation: accept after minor revisions

This is a revised manuscript that I reviewed before. I still have some comments on the manuscript, see below. Also, some of my previous comments were left unanswered. For instance, the details of the coupling or a suggestion to modify the figures so GCM results are better distinguished from SP-GCM.

**Specific (mostly minor) comments.**

1. Long sentences are difficult to follow in several (many?) places. This is mostly because of missing commas in long sentences. I understand that the authors are not native speakers, but a professional help would be needed if the journal does not provide technical editing. Similarly, there are some awkward sentences/statements that need to be corrected (e.g., P9L7: "too less", maybe "not enough"?).

2. Following on my comment 5 from the previous review: which fields are coupled between GCM and CRM? I think temperature and moisture for sure. What about momenta? Vertical momentum cannot be coupled as it has to be zero in the horizontally-periodic CRM. How are the horizontal momenta coupled? Also, I assume that the large-scale moist physics (cloud and precipitation formation, precipitation fallout, etc.) is inactive in SP simulations. Are mean cloud and precipitation fields advected by the host GCM? Please expand the discussion of the SP GCM formulation.

3. The most outstanding feature of simulations vs observations in Table 2 is the factor 3 to 4 overestimation of the LWP and a significant overestimation of IWP. These are briefly discussed in the text, but I feel they should be strongly emphasized in the discussion, including possible reasons.

4. I feel the authors should strongly emphasize that the traditional GCM simulation, CTRL, is the outcome of a significant tuning and years of experience of the model developers. This is in contrast to virtually no tuning in SP-GCM.

5. It is not clear from Table 2 and equation on p. 7 that the NetCRE is at the top of the atmosphere.

6. I feel a better introduction to the Taylor diagram would help the reader.

7. This is comment 15 from the previous review: Figure 2 (and maybe other figures). I suggested not to use a color for CTRL, but a symbol (e.g., a star). The authors claim this is not possible. I do not agree, but leave it to the authors to decide. Another possibility would be to use the same symbol (circle, square, etc.), and the white color inside. The key point is that CTRL should be more distinct from other simulation results.

8. P. 10. Some of the statements (e.g., L16-18, 48-52) seem not supported by the figures. Please clarify.

9. Figure 5. First, the line colors correspond to the colors above the inserted panels, correct? If so, please state that in the caption. The Warm Pool inserts show the relatively clearly the geography, but this is unclear in the inserted panels on the right. Does the narrow vertical white strip represent South America? Would it help if the lat-lon scale is preserved between the inserted panels? That is, inserted panels on the

right should be narrow strips with different aspect ratio than the left panels. ("Sight" in the caption should be "side").

10. P. 15. The list in the right column is unclear. First, SP-EMAC is not needed. OR1 lists east-west orientation, but OR2 and OR3 exclude that information.

11. Figure 10. First, the captions above lower 9 panels are barely visible. Please make them much larger and bold, perhaps within the panels. The most outstanding feature of this figure is that panels across the diagonals are similar. That is, the upper left panel looks similar to the lower right panel, and the lower left is similar to the upper right. This should be noticed upfront and possible reasons can be included in the discussion.

12. Section 4. I feel some of the specific results summarized here agree with previous studies. It would be good to include more references in this section.

---

## Author Response (AR2)

**Authors response to second review:**

Review comments pasted in black.
Author response in blue.

We would like to thank the reviewers again for the helpful report. The manuscript has been revised according to the referees comments.

**Review No. 1**
General comments:
This revised manuscript focuses the diversity of precipitation and cloud optical thickness of the superparameterized model, by fixing the bug for the simulations in the first submission. The results became more reasonable and the message of the paper became clearer. I suggest publication of this paper after fixing the points indicated below.

Specific comments:
p. 9, L28, 'partition': It is unclear what kind of partition is stated here.
'partition' is replaced by 'distribution'.

p. 11, L41, 'TRMM data': More explicitly refer to the TRMM product name as 'TRMM_3B42 v7'. In Sato et al. (2009,JCLI), the distinction between 3B42 and 3G68 is argued, and it is stated that 3G68 is more reliable for the diurnal cycle.
The explicit product name has been used! Thank you for mentioning the study of Sato et al. (JCLI,2009) and references therein. We have mentioned the phase lag of approx. 3 hours in 3B42 compared to 3G68. The cause of this time lag may be related to infrared precipitation estimates which tends to lag behind in situ observations Kikuchi and Wang (JCLI,2008). We have added a little bit more discussion about the timing of the different models, taking into account that TRMM_3B42 has a time lag.

p. 15, Figure 9: In this figure, the Taylor diagram of two quantities are shown: sensible and latent heat fluxes and cloud optical thickness (COT). These two quantities are not related. The left panel (sensible and latent heat fluxes) can be removed. Instead, it would be more informative if some relation exists between precipitation and COT.
It is correct that these quantities are not related. In our opinion it is important to support the results of older literature which still holds for various CRM configurations. Additionally, it is essential to mention that some aspects are hardly influenced by changing configuration of the superparameterisation. We would like to display this figure as it is.

p. 18, L5-9, 'Nowadays state-of-the-art global cloud resolving models provide new possibilities comparing superparameterised simulations with monthly-long high resolution models (Stevens et al., 2019).': This sentence is not suitable for this paragraph. This should be moved to the introduction.
The paragraph has been moved to the introduction.

**Review No. 2** General comments:

This is a revised manuscript that I reviewed before. I still have some comments on the manuscript, see below. Also, some of my previous comments were left unanswered. For instance, the details of the coupling or a suggestion to modify the figures so GCM results are better distinguished from SP-GCM.

1. Long sentences are difficult to follow in several (many?) places. This is mostly because of missing commas in long sentences. I understand that the authors are not native speakers, but a professional help would be needed if the journal does not provide technical editing. Similarly, there are some awkward sentences/statements that need to be corrected (e.g., P9L7: "too less", maybe "not enough"?).

We thank the reviewer for the comment and edited the manuscript focusing on long sentences/missing commas!

2. Following on my comment 5 from the previous review: which fields are coupled between GCM and CRM? I think temperature and moisture for sure. What about momenta? Vertical momentum cannot be coupled as it has to be zero in the horizontally-periodic CRM. How are the horizontal momenta coupled? Also, I assume that the large-scale moist physics (cloud and precipitation formation, precipitation fallout, etc.) is inactive in SP simulations. Are mean cloud and precipitation fields advected by the host GCM? Please expand the discussion of the SP GCM formulation.

The description of the GCM-CRM coupling has been expanded. Fields that are fully (GCM forcing and CRM feedback) coupled are: temperature and moisture (water vapor, cloud water, cloud ice). Horizontal momenta (u- and v-wind component) are fully coupled for full 3D CRM simulations. Concerning the 2D-oriented versions of SP-EMAC, the large-scale wind (u component for OR1 and v component in the case of OR2) forces the CRM but no small-scale (CRM) feedback is allowed. Vertical momentum is not coupled at all, as stated in the comment.
Large-scale moist physics are turned off when selecting the superparameterisation. Advection of mean (average over all CRM cells within a single GCM column) prognostic fields (moisture, temperature, vorticity, etc.) is active, this ensures propagation of large-scale phenomena from one GCM grid cell to neighboring cells. Precipitation or cloud cover is not advected since these are diagnostic variables. It is assumed that hydrometeors are removed/sedimented within a GCM time step.

3. The most outstanding feature of simulations vs observations in Table 2 is the factor 3 to 4 overestimation of the LWP and a significant overestimation of IWP. These are briefly discussed in the text, but I feel they should be strongly emphasized in the discussion, including possible reasons.

We have discussed this issue in more detail and provided several reasons how the overestimation of IWP is connected to the cloud microphyiscs within the CRM.

4. I feel the authors should strongly emphasize that the traditional GCM simulation, CTRL, is the outcome of a significant tuning and years of experience of the model developers. This is in contrast to virtually no tuning in SP-GCM.

It has been mentioned several times within the manuscript that almost no tuning has been performed for SP-EMAC. More effort is needed to complete the tuning process for SP-EMAC. An additional sentence, that many tuning activities for CTRL have been performed, has been added in the discussion.

5. It is not clear from Table 2 and equation on p. 7 that the NetCRE is at the top of the atmosphere.

At the beginning of chapter 3.1 it has been mentioned that the net radiative flux and cloud radiative effect are related to the TOA. Additional information is now given in Table 2 (caption) and above the equation on p.7.

6. I feel a better introduction to the Taylor diagram would help the reader.

The short introduction to the Taylor diagram has been expanded. For further details the reader is referred to the literature of Karl E. Taylor.

7. This is comment 15 from the previous review: Figure 2 (and maybe other figures). I suggested not to use a color for CTRL, but a symbol (e.g., a star). The authors claim this is not possible. I do not agree, but leave it to the authors to decide. Another possibility would be to use the same symbol (circle, square, etc.), and the white color inside. The key point is that CTRL should be more distinct from other simulation results.

The symbol(s) for the control simulation (for all Taylor diagrams) have been changed to be more distinct from all other simulations. We have still used purple colored, but open symbols with an additional white color inside.

8. P. 10. Some of the statements (e.g., L16-18, 48-52) seem not supported by the figures. Please clarify.

P.10: L16-18: This statement cannot be deduced from the figure. An analysis of all individual simulations reveals the sensitivity for these regions, which is not shown in the manuscript but is important to notice within this section.

P.10: L48-52: Again, this statement can be verified if all individual lines for all SP-EMAC simulations are plotted in figure 5. This has not been done, because it would have looked too messy.

9. Figure 5. First, the line colors correspond to the colors above the inserted panels, correct? If so, please state that in the caption. The Warm Pool inserts show the relatively clearly the geography, but this is unclear in the inserted panels on the right. Does the narrow vertical white strip represent South America? Would it help if the lat-lon scale is preserved between the inserted panels? That is, inserted panels on the right should be narrow strips with different aspect ratio than the left panels. ("Sight" in the caption should be "side").

The caption for figure 5 has been changed to include additional information about inserted maps and corresponding colors. The inserted panels show maps including the specified geographical region given in the text.

10. P. 15. The list in the right column is unclear. First, SP-EMAC is not needed. OR1 lists east-west orientation, but OR2 and OR3 exclude that information.
'SP-EMAC' removed. OR2 and OR3 now include the information about the CRM orientation.

11. Figure 10. First, the captions above lower 9 panels are barely visible. Please make them much larger and bold, perhaps within the panels. The most outstanding feature of this figure is that panels across the diagonals are similar. That is, the upper left panel looks similar to the lower right panel, and the lower left is similar to the upper right. This should be noticed upfront and possible reasons can be included in the discussion.
Top captions of the 9 panels have been edited. A short notice upon the similarity of the diagonal panels and possible reasons have been added.

12. Section 4. I feel some of the specific results summarized here agree with previous studies. It would be good to include more references in this section.
Additional references have been added to the last section.

[revised manuscript text omitted]